# iCITRIS: Causal Representation Learning for Instantaneous Temporal Effects

**Phillip Lippe**[1]    **Sara Magliacane**[2,3]    **Sindy Löwe**[4]    **Yuki M. Asano**[1]    **Taco Cohen**[5]    **Efstratios Gavves**[1]

[1]QUVA Lab, University of Amsterdam
[2]MIT-IBM Watson AI Lab
[3]INDE Lab, University of Amsterdam
[4]UvA-Bosch Delta Lab, University of Amsterdam
[5] Qualcomm AI Research*, Amsterdam, Netherlands

## Abstract

Causal representation learning is the task of identifying the underlying causal variables and their relations from high-dimensional observations, such as images. Recent work has shown that one can reconstruct the causal variables from temporal sequences of observations under the assumption that there are no instantaneous causal relations between them. In practical applications, however, our measurement or frame rate might be slower than many of the causal effects. This effectively creates "instantaneous" effects and invalidates previous identifiability results. To address this issue, we propose iCITRIS, a causal representation learning method that can handle instantaneous effects in temporal sequences when given perfect interventions with known intervention targets. iCITRIS identifies the intervention-dependent part of the causal factors from temporal observations, while simultaneously using a differentiable causal discovery method to learn their causal graph. We demonstrate this in experiments on two video datasets.

## 1 INTRODUCTION

Causal representation learning aims at learning representations of causal factors in an underlying system from high-dimensional observations like images (Brehmer et al., 2022; Locatello et al., 2020a; Schölkopf et al., 2021; Yang et al., 2021). Several works have considered identifying causal factors from time series data, assuming that the variables are independent of each other conditioned on the previous time step (Gresele et al., 2021; Khemakhem et al., 2020; Lachapelle et al., 2022; Lippe et al., 2022b; Yao et al., 2022). This scenario assumes that within each discrete, measured time step, intervening on one causal factor does not affect

any other instantaneously. However, in real-world systems, this assumption is often violated, as there might be causal effects that act faster than the measurement or frame rate. For instance, consider the example of a light switch and a light bulb. When flipping the switch, there is an almost immediate effect on the light by turning it on or off. In this case, an intervention on a variable (*e.g.* the switch) also affects other variables (*e.g.* the bulb) in the same time step, violating the assumption that each variable is independent of the others in the same time step, conditioned on the previous time step.

To overcome this limitation, we consider the task of identifying causal variables and their causal graphs from temporal sequences, even under potentially instantaneous cause-effect relations. This task contains two main challenges: disentangling the causal factors from observations, and learning the causal relations between those factors. As opposed to temporal sequences without instantaneous effects, neither of these two tasks can be completed without the other: without knowing the variables, we cannot identify the graph; but without knowing the graph, we cannot disentangle the causal variables since they are not conditionally independent. In particular, in contrast to causal relations across time steps, the orientations of instantaneous edges are not determined by the temporal ordering, hence requiring to jointly solve the task of causal representation learning and causal discovery.

As a first step, we show that in the presence of potential instantaneous causal effects, we cannot identify the causal factors from observations, unless we have access to perfect interventions or strong additional assumptions. Intuitively, if the graph remains unchanged in experiments, one cannot distinguish between entanglements in the observational space (*e.g.* images) and instantaneous causal relations. For example, consider a blue object illuminated by a white spotlight. If we change the spotlight color to red, the object will appear black. From purely observational data, we cannot distinguish whether the spotlight's color caused a change in the object's actual color, or if it just changed its appearance. However, if after performing a perfect intervention on the object color, *e.g.* by fixing it to green, we observe that the perceived

---

* Qualcomm AI Research is an initiative of Qualcomm Technologies, Inc.

*Accepted for the Causal Representation Learning workshop at the 38th Conference on Uncertainty in Artificial Intelligence* (UAI CRL 2022).

object color is not the one we expected, then we can deduce how the entanglement happens in the observation function.

Further, we consider the general setting in which causal factors can be multidimensional. Following Lippe et al. (2022b), we focus on the *minimal causal variables*, *i.e.* the parts of the causal factors that are affected by the interventions, since interventions may leave some dimensions unchanged. In this setting, we prove that we can identify the minimal causal variables and their graph, if we have sequences with perfect interventions on known targets.

As a practical implementation, we propose *instantaneous CITRIS*, or iCITRIS, which, inspired by the recent causal representation learning method CITRIS (Lippe et al., 2022b), can discover the minimal causal variables and their causal graph for both instantaneous and temporal effects. iCITRIS maps high-dimensional observations like images to a latent space, on which it learns an instantaneous causal graph by integrating a causal discovery method into its prior. In particular, we consider two recent differentiable causal discovery methods: NOTEARS (Zheng et al., 2018) and ENCO (Lippe et al., 2022a). In experiments on two video datasets, we show that iCITRIS can disentangle the causal variables while accurately recovering their causal graph.

**Related work** Most works in the field of causal representation learning have focused so far on identifying independent factors of variations from data (Klindt et al., 2021; Kumar et al., 2018; Locatello et al., 2019, 2020b; Träuble et al., 2021), including recent works in Independent Component Analysis (ICA) (Gresele et al., 2021; Hyvärinen et al., 2001, 2019; Monti et al., 2019). In particular, Lachapelle et al. (2022); Yao et al. (2022) discuss identifiability of causal variables from temporal sequences, but require all causal variables to be conditionally independent and scalar. Focusing on causal structures in the data, von Kügelgen et al. (2021) demonstrate that contrastive learning methods can block-identify causal variables invariant to augmentations. CITRIS (Lippe et al., 2022b) uses temporal sequences with interventions to identify the minimal causal variables, *i.e.* the part of a potentially multidimensional causal variable that is influenced by the provided interventions. Still, similar to works on ICA, CITRIS requires the causal variables within a time step to be independent conditioned on the previous time step, which is violated by instantaneous effects. Locatello et al. (2020a) identify independent latent causal factors from pairs of observations that only differ in a subset of causal factors. Brehmer et al. (2022) have recently extended this setup to variables with instantaneous causal effects. However, in contrast to our approach, Brehmer et al. (2022) require counterfactual observations where for a pair of observations, the noise term for all variables is identical, except for a single intervened variable. To the best of our knowledge, iCITRIS is the first method to identify causal variables and their causal graph from temporal, intervened sequences in the case of potentially instantaneous causal effects.

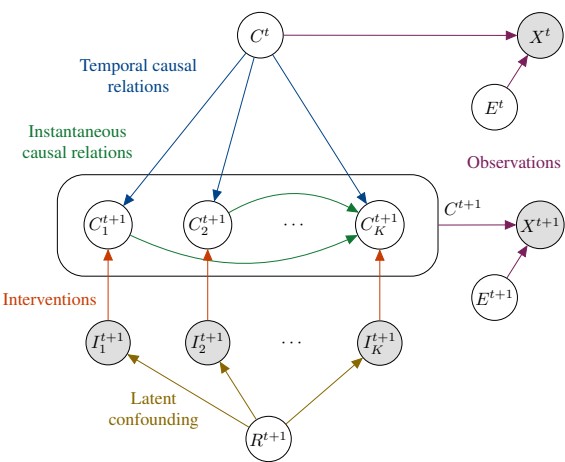

Figure 1: An example causal graph in iTRIS. A latent causal factor $C_i^{t+1}$ can have as potential parents the causal factors at the previous time step $C^t = (C_1^t, \ldots, C_K^t)$, instantaneous parents $C_j^{t+1}, i \neq j$, and its intervention target $I_i^{t+1}$. All causal variables $C^{t+1}$ and the noise $E^{t+1}$ cause the observation $X^{t+1}$. $R^{t+1}$ is a potential latent confounder between the intervention targets.

## 2 IDENTIFIABILITY IN TEMPORAL INTERVENED SEQUENCES WITH INSTANTANEOUS EFFECTS

We first describe our setting, Instantaneous TempoRal Intervened Sequences (iTRIS). We then discuss the challenges that arise due to instantaneous effects, and their implications on the identifiability of the causal factors. Finally, we present the identifiability results in iTRIS.

### 2.1 INSTANTANEOUS TEMPORAL INTERVENED SEQUENCES (ITRIS)

iTRIS considers a latent temporal causal process with $K$ *causal factors* $(C_1^t, C_2^t, ..., C_K^t)_{t=1}^T$ with both causal relations across time steps, *i.e.* temporal, and within a time step, *i.e.* instantaneous. At each time step $t$, we measure a high-dimensional observation $X^t$ from this process, representing a noisy, entangled view of all causal factors $C^t = (C_1^t, C_2^t, ..., C_K^t)$. The following paragraphs describe this setup in more detail, which is visualized in Figure 1.

**Causal factors:** We consider causal factors to be potentially multidimensional, *i.e.*, $C_i \in \mathcal{D}_i^{M_i}$ with $M_i \geq 1$ being the unobserved dimensionality of $C_i$ and $\mathcal{D}_i$ the domain, for example $\mathbb{R}$ for continuous variables. We define the causal factor space as $\mathcal{C} = \mathcal{D}_1^{M_1} \times ... \times \mathcal{D}_K^{M_K}$.

**Causal structure assumptions:** We assume that the underlying latent causal process is a dynamic Bayesian network (DBN) (Dean and Kanazawa, 1989; Murphy, 2002) over the multidimensional random variables $(C_1, C_2, ..., C_K)$ that is first-order Markov, stationary, and potentially has instanta-

neous effects between different variables. This means that each causal factor $C_i$ is instantiated at each time step $t$, denoted by $C_i^t$, and its causal parents $\mathrm{pa}(C_i^t)$ can be any subset of the causal factors at time $t-1$ and $t$, including its own previous value $C_i^{t-1}$. Additionally, we assume that the complete structure of the graph is acyclic. Furthermore, the graph and its parameters are assumed to be time-invariant (*i.e.*, they repeat across each time step). Finally, we assume that the causal factors are causally sufficient (*i.e.*, there are no additional latent confounders) and that the distribution satisfies the causal faithfulness assumption (*i.e.*, there are no additional independences w.r.t. the ones encoded in the graph). Further details are provided in Appendix A.1.

**Interventions:** In each time step, we consider that interventions may have been performed on a subset of causal factors. We assume that we have access to the intervention targets, but not the sampled values of the intervened variables. We denote these targets by the binary vector $I^{t+1} \in \{0,1\}^K$ where $I_i^{t+1} = 1$ refers to an intervention that has been performed on the causal variable $C_i^{t+1}$. Further, we assume the observational and interventional distributions to share the same support on $\mathcal{C}$. Finally, the intervention targets may be confounded by a latent regime variable $R^{t+1}$.

**Observation function:** We define the observation function $h(C_1^t, C_2^t, ..., C_K^t, E^t) = X^t$, where $E^t$ is any noise affecting the observation $X^t$ independent of the causal factors $C^t$, and $h : \mathcal{C} \times \mathcal{E} \to \mathcal{X}$ is a function from the causal factor space $\mathcal{C}$ and the space of the noise variables $\mathcal{E}$ to the observation space $\mathcal{X}$. We assume that $h$ is bijective, so we can identify each causal factor uniquely from observations.

## 2.2 CHALLENGES OF INSTANTANEOUS CAUSAL EFFECTS

In iTRIS, causal variables within a time step can be related in two ways: they can be entangled by the observation function $h$ to create the observation $X^t$, or they are causally related in the instantaneous causal graph, as discussed in the motivating example with a spotlight and an object changing colors in the introduction. To truly disentangle and identify the causal variables, we need to distinguish between these two forms of entanglement. Intuitively, a perfect intervention on a variable removes all its incoming edges, including instantaneous effects, thus excluding one type of entanglement.

To formalize this intuition, we consider an example in iTRIS with only two latent causal factors $C_1$ and $C_2$, and, for simplicity, without temporal relations. We assume $C_1$ and $C_2$ do not cause each other, and the observation function is simply $X = [C_1, C_1 + C_2]$. We show that one cannot identify the causal factors or their graph from $p(X)$, since there are multiple different representations that can model $p(X)$ equally well. For instance, an alternative representation is $\hat{C}_1 = X_1 = C_1, \hat{C}_2 = X_2 = C_1 + C_2$ with the causal graph

$\hat{C}_1 \to \hat{C}_2$, since $p(\hat{C}_2|\hat{C}_1) = p(C_1 + C_2|C_1) = p(C_2)$ and hence $p(\hat{C}_1, \hat{C}_2) = p(C_1, C_2) = p(X)$. Even under soft interventions that change the conditional distributions of $C_1$ and $C_2$, we cannot necessarily tell the two representations apart, because the causal graph can remain unchanged.

Perfect interventions, on the other hand, force the intervened variable to be independent of its parents, which includes instantaneous effects. Thus, under perfect interventions on $C_2$, the identified representation of $C_2$ must be independent of all its parents, including any potential instantaneous effect. This eliminates the possible representation of $\hat{C}_2 = C_1 + C_2$, since $C_1 + C_2 \not\perp C_1 | I_2 = 1$ implies $\hat{C}_2 \not\perp \hat{C}_1 | I_2 = 1$. Thus, we come to the following conclusion:

**Lemma 2.1.** *A causal variable $C_i$ cannot always be uniquely identified in iTRIS if $C_i$ potentially has instantaneous parents and the available data does not contain any perfect intervention on $C_i$.*

We provide the proof for this lemma and an example with temporal relations in Appendix A.2.1. This lemma implies that without perfects interventions on a causal variable $C_i$, we can only disentangle it from a set of other causal variables, $\tilde{C}$ where $C_i \notin \tilde{C}$, if we assume that $C_i$ does not have instantaneous parents in $\tilde{C}$. Otherwise, there may exist additional representations, similar to the example before, where $C_i$ is entangled with other causal variables in $\tilde{C}$.

## 2.3 IDENTIFYING THE MINIMAL CAUSAL VARIABLES

As a result of Lemma 2.1, we extend our assumptions to identify the causal variables. First, we assume that all interventions performed are perfect, *i.e.* $p(C_i^t|\mathrm{pa}(C_i^t), I_i^t = 1) = p(C_i^t|I_i^t = 1)$. We require that the causal variables on which interventions are never performed in the observed sequence, *i.e.* $\forall t, I_i^t = 0$, are not instantaneous children of any variable with observed interventions. Further, to differentiate between interventions on different variables, we assume that no intervention target is a deterministic function of any other, *e.g.* by not allowing $\forall t, I_i^t = I_j^t, i \neq j$. Finally, if there exist symmetries in the distribution of a causal variable $C_i$, the representation of $C_i$ may depend on other causal variables without changing the likelihood. For instance, a one dimensional Gaussian has the symmetry of flipping the variable $C_i$ around its mean. Thus, $C_i$ could be flipped, conditioned on another variable $C_j, i \neq j$, without changing the likelihood. To break this symmetry, the mean must depend on the intervention target and/or any variable from the previous time step. Hence, we assume that the temporal dependencies and provided interventions break all symmetries in the distributions. Appendix A.2 provides further details on the assumptions and their necessity.

Under these assumptions, we aim to identify the causal variables from high-dimensional observations. As shown by

Lippe et al. (2022b), multidimensional causal variables are not always identifiable, since interventions can potentially only affect a subset of the variables dimensions. Thus, we are limited to identifying the intervention-dependent parts of variables, *i.e.* the *minimal causal variables*.

In particular, we learn an invertible mapping, $g_\theta : \mathcal{X} \rightarrow \mathcal{Z}$, from observations $\mathcal{X}$ to a latent space $\mathcal{Z} \in \mathbb{R}^M$ with $M$ dimensions. The latent space is structured by an assignment function $\psi : [\![1..M]\!] \rightarrow [\![0..K]\!]$ mapping each dimension of $\mathcal{Z}$ to a causal factor $C_1, ..., C_K$. We index the set of latent variables that $\psi$ assigns to the causal factor $C_i$ with $z_{\psi_i} = \{z_j | j \in [\![1..M]\!], \psi(j) = i\}$. Thereby, we use $z_{\psi_0}$ to summarize all latents that model the noise $E_o^t$ and the intervention-independent parts of all causal variables. Furthermore, to model the instantaneous causal relations, we learn a directed, acyclic graph $G$ on the $K$ latent variable groups $z_{\psi_0}, ..., z_{\psi_K}$. The graph $G$ induces a parent structure denoted by $z_{\psi_i^{pa}} = \{z_j | j \in [\![1..M]\!], \psi(j) \in \mathrm{pa}_G(i)\}$ where we set $\mathrm{pa}_G(0) = \emptyset$, *i.e.* the variables in $z_{\psi_0}$ have no instantaneous parents. Meanwhile, the temporal causal graph, *i.e.* between $C^t$ and $C^{t+1}$, is implicitly learned by conditioning the latent variables of a time step, $z^{t+1}$, on all latents of the previous time step, $z^t$. Since the orientations of the edges are known, *i.e.* from $C^t$ to $C^{t+1}$, no causal discovery setup as for the instantaneous graph is strictly necessary, and the temporal graph can instead be pruned in a post-processing step. Overall, this results in the following prior:

$$p_{\phi,G}\left(z^{t+1}|z^t, I^{t+1}\right) = p_\phi\left(z_{\psi_0}^{t+1}|z^t\right) \cdot$$
$$\prod_{i=1}^{K} p_\phi\left(z_{\psi_i}^{t+1}|z^t, z_{\psi_i^{pa}}^{t+1}, I_i^{t+1}\right) \quad (1)$$

Under interventions, we mask out the parents $z^t$ and $z_{\psi_i^{pa}}^{t+1}$ for the prior of $z_{\psi_i}^{t+1}$, maintaining the independence relations under perfect interventions. Given a dataset of triplets $\{x^t, x^{t+1}, I^{t+1}\}$ with observations $x^t, x^{t+1} \in \mathcal{X}$ and intervention targets $I^{t+1}$, the full likelihood objective becomes:

$$p_{\phi,\theta,G}(x^{t+1}|x^t, I^{t+1}) = \left|\det J_{g_\theta}(x^{t+1})\right| p_{\phi,G}(z^{t+1}|z^t, I^{t+1}) \quad (2)$$

where the Jacobian of the invertible map $g_\theta$, $\left|\det J_{g_\theta}(x^{t+1})\right|$, comes from the change of variables of $x$ to $z$. Under this setup, we derive the following result:

**Theorem 2.2.** *Suppose that $\phi^*, \theta^*, \psi^*$ and $G^*$ are the parameters that, under the constraint of maximizing the likelihood $p_{\phi,\theta,G}(x^{t+1}|x^t, I^{t+1})$, maximize the information content of $p_\phi(z_{\psi_0}^{t+1}|z^t)$ and minimize the edges in $G^*$. Then, with sufficient latent dimensions, the model $\phi^*, \theta^*, \psi^*$ learns a latent structure where $z_{\psi_i}^{t+1}$ models the minimal causal variable of $C_i$, and $G^*$ is the true instantaneous graph between these variables. Removing edges based on conditional independencies between time steps $t$ and $t+1$ identifies the true temporal graph. Finally, $z_{\psi_0}$ models all remaining information.*

The proof for this theorem in Appendix A follows three main steps. First, we show that the true disentanglement function constitutes a global optimum of the likelihood objective of Equation (2), but is not necessarily unique. Second, we derive that any global optimum must have disentangled the minimal causal variables in the latent representation, and that maximizing the information content of $z_{\psi_0}$ uniquely identifies the minimal causal variables. Finally, we show that under this disentanglement, optimizing the likelihood of the observational and interventional data identifies the complete causal graph between the minimal causal variables.

Intuitively, this theorem shows that we can identify and disentangle the minimal causal variables, even when instantaneous effects are present. Furthermore, we are able to find the instantaneous causal graph $G^*$ between the minimal causal variables, which, however, might not be exactly the same as the causal graph $G_C$ between the true causal variables $C_1^t, ..., C_K^t$. Every edge $z_{\psi_i}^t \rightarrow z_{\psi_j}^t$ in $G^*$ implies an edge $C_i^t \rightarrow C_j^t$ in $G_C$. However, some edges in $G_C$ might be modeled by relations between intervention-independent parts of causal variables, which are captured by an edge $z_{\psi_0}^t \rightarrow z_{\psi_j}^t$ in $G^*$. Still, translating the edges from $G^*$ to $G_C$ following the previous implication does not introduce any wrong edge or orientation that was not present in $G_C$.

## 3 LEARNING VARIABLES WITH INSTANTANEOUS EFFECTS

Based on the theoretical results presented above, we propose iCITRIS, a causal representation learning method that simultaneously identifies the causal variables and the instantaneous causal graph between them. Inspired by CITRIS (Lippe et al., 2022b), iCITRIS implements the mapping from observations to latent space either by a variational autoencoder (VAE) (Kingma and Welling, 2014) or by a normalizing flow (Rezende and Mohamed, 2015) trained on the representations of a pretrained autoencoder. In both cases, the disentanglement of the causal variables in latent space is promoted by enforcing the structure of Equation (1). However, crucially, the instantaneous graph $G^*$ must be learned jointly with the causal representations, as we describe next.

### 3.1 LEARNING THE INSTANTANEOUS GRAPH

To learn the instantaneous causal graph simultaneously with the causal representation, we incorporate recent differentiable, score-based causal discovery methods in iCITRIS. Given a prior distribution over graphs $p(G)$, the prior distribution over the latent variables $z^{t+1}$ of Equation (1) is:

$$p_\phi\left(z^{t+1}|z^t, I^{t+1}\right) = p_\phi\left(z_{\psi_0}^{t+1}|z^t\right) \cdot$$
$$\mathbb{E}_G\left[\prod_{i=1}^{K} p_\phi\left(z_{\psi_i}^{t+1}|z^t, z_{\psi_i^{pa}}^{t+1}, I_i^{t+1}\right)\right] \quad (3)$$

where the parent sets, $z_{\psi_i^{pa}}^{t+1}$, depend on the graph structure $G$. The goal is to jointly optimize $p_\phi$ and $p(G)$ under maximizing the likelihood objective of Equation (2), such that $p(G)$ is peaked at the correct causal graph. To apply causal discovery methods in this setup, we consider each group of latents, $z_{\psi_i}^{t+1}$, as the potentially multidimensional causal factor $C_i$, on which the graphs must be recovered. To this end, we experiment with two different discovery methods: NOTEARS (Zheng et al., 2018), and ENCO (Lippe et al., 2022a).

**NOTEARS** (Zheng et al., 2018) casts structure learning as a continuous optimization problem by providing a continuous constraint to enforce acyclicity. Specifically, an adjacency matrix $A$ is acyclic if the following holds: $c(A) = \text{tr}\left(\exp(A \circ A)\right) - K = 0$, where $\circ$ is the Hadamard product, $\exp(...)$ the matrix exponential, and $K$ the number of causal variables. Following Ng et al. (2022), we model the adjacency matrix $A$ with independent edge likelihoods, and differentially sample from it using the Gumbel-Softmax trick (Jang et al., 2017). We use these samples as graphs in the prior $p_\phi\left(z^{t+1}|z^t, I^{t+1}\right)$ to mask the parents of the individual causal variables, and obtain gradients for the graph through the maximum likelihood objective of the prior. In order to promote acyclicity, we use the constraint $c(A)$ as a regularizer, and exponentially increase its weight over training. This ensures that at the end of the training, $c(A)$ is close to zero, and, thus, the predicted graph is acyclic.

**ENCO** (Lippe et al., 2022a), on the other hand, uses interventional data and a different graph parameterization to obtain acyclic graphs. The graph parameters are split into two sets: one for modeling the orientation per edge, and one for whether the edge exists or not. By using solely adjacent interventions to update the orientation parameters, Lippe et al. (2022a) show that ENCO naturally converges to the true, acyclic graph in the sample limit. Since iCITRIS requires interventions on variables that have potential edges of unknown orientations, we can integrate ENCO without additional constraints on the dataset. Instead of the Gumbel-Softmax trick, ENCO uses low-variance, unbiased gradients based on REINFORCE (Williams, 1992) to update the graph parameters, potentially providing a more stable optimization than NOTEARS. For efficiency, we merge the distribution and graph learning stage of ENCO, and update both the graph and distribution parameters at each iteration.

### 3.2 STABILIZING THE OPTIMIZATION PROCESS

The main challenge in iCITRIS is that simultaneously identifying the causal variables and their graph leads to a chicken-and-egg situation: without knowing the variables, we cannot identify the graph; but without knowing the graph, we cannot disentangle the causal variables. This can cause the optimization to be naturally unstable and to converge to local minima with incorrect graphs. To stabilize the optimization process, we propose the following two approaches.

**Graph learning scheduler** During the first training iterations, the assignment of latent to causal variables is almost uniform, such that the gradients for the graph parameters are very noisy and uninformative. Thus, we use a learning rate schedule for the graph learning parameters such that the graph parameters are frozen for the first couple of epochs. During those training iterations, the model learns to fit the latent variables to the intervention variables under an arbitrary graph, leading to an initial, rough assignment of latent to causal variables. Then, we warm up the learning rate to slowly start the graph learning process while continuing to disentangle the causal variables in latent space.

**Mutual information estimator** Since iCITRIS requires perfect interventions, we can further exploit this information in the optimization process by enforcing independencies between parents and children under interventions. In particular, under interventions on the variable $C_i$ at time $t$, *i.e.* $I_i^t = 1$, the following independencies must hold: $C_i^t \perp\!\!\!\perp \text{pa}(C_i^t)|I_i^t = 1$. The same independencies can be transferred to the latent space as $z_{\psi_i}^t \perp\!\!\!\perp (z_{\psi_i^{pa}}^t, z^{t-1})|I_i^t = 1$. To use these independencies as learning signals in iCITRIS's gradient-based framework, we use the fact that two variables being independent corresponds to the mutual information (MI) of both to be zero (Kullback, 1997). Following previous work on neural network based MI estimation (Belghazi et al., 2018; Hjelm et al., 2019; van den Oord et al., 2018), we implement this framework by training a network to distinguish between samples from the joint distribution $p(z_{\psi_i}^t, z_{\psi_i^{pa}}^t, z^{t-1}|I_i^t = 1)$ and the product of their marginals, i.e. $p(z_{\psi_i}^t|I_i^t = 1)p(z_{\psi_i^{pa}}^t, z^{t-1}|I_i^t = 1)$. While the MI estimator network is trained to optimize its classification accuracy, the latents are optimized to do the opposite, effectively minimizing the mutual information between $z_{\psi_i}^t$ and its parents under interventions.

## 4 EXPERIMENTS

We evaluate iCITRIS on two videos datasets and compare it to common disentanglement methods. We publish our code at `https://github.com/phlippe/CITRIS`.

### 4.1 EXPERIMENTAL SETTINGS

**Baselines** Since iCITRIS is, to the best of our knowledge, the first method to identify causal variables with instantaneous effects in this setting, we compare it to methods for disentangling conditionally independent causal variables. Firstly, we use CITRIS (Lippe et al., 2022b) and the Identifiable VAE (iVAE) (Khemakhem et al., 2020), which both assume that the variables are independent given the previous time step and intervention targets. Additionally, to also compare to a model with dependencies among latent variables, we evaluate the iVAE with an autoregressive prior,

Table 1: Results on Instantaneous Temporal Causal3DIdent over three seeds. iCITRIS-ENCO performs best in identifying the variables and their graph.

| Model | $R^2$ (diag ↑ / sep ↓) | SHD (instant ↓ / temp ↓) |
|---|---|---|
| iCITRIS-ENCO | **0.96** / **0.05** | **1.33** / **5.00** |
| iCITRIS-NOTEARS | 0.95 / 0.09 | 4.00 / **5.00** |
| CITRIS | 0.92 / 0.19 | 4.67 / 10.00 |
| iVAE | 0.82 / 0.20 | 6.67 / 15.33 |
| iVAE-AR | 0.79 / 0.29 | 11.00 / 12.67 |

Table 2: Results on the Causal Pinball dataset (three seeds).

| Model | $R^2$ (diag ↑ / sep ↓) | SHD (instant ↓ / temp ↓) |
|---|---|---|
| iCITRIS-ENCO | **0.98** / **0.04** | **0.67** / **3.67** |
| iCITRIS-NOTEARS | **0.98** / 0.06 | 2.33 / **3.67** |
| CITRIS | **0.98** / **0.04** | 2.67 / 4.00 |
| iVAE | 0.55 / **0.04** | 2.33 / 4.33 |
| iVAE-AR | 0.53 / 0.15 | 4.33 / 6.33 |

which we denote with iVAE-AR. To ensure comparability, we share the general model setup where possible (*e.g.* encoder/decoder network) across all methods.

**Evaluation metrics** We follow Lippe et al. (2022b) in reporting the $R^2$ correlation scores between the true causal factors and the latent variables that have been assigned to a specific causal variable by the learned model. We denote the average correlation of the predicted causal factor to its true value with $R^2$-diag (optimal 1), and the maximum correlation besides its true factor with $R^2$-sep (optimal 0). Furthermore, to investigate the modeling of the temporal and instantaneous relations between the causal factors, we perform causal discovery as a post-processing step on the latent representations of all models, and report the structural hamming distance (SHD) between the predicted and true causal graph.

### 4.2 3D OBJECT RENDERINGS: CAUSAL3DIDENT

We use the Temporal Causal3DIdent dataset (von Kügelgen et al., 2021; Lippe et al., 2022b) which contains 3D renderings ($64 \times 64$ pixels) of different object shapes under varying positions, rotations, and lights. To introduce instantaneous effects into the dataset, we replace all temporal relations with instantaneous edges, except those on the same variable ($C_i^t \to C_i^{t+1}$). For instance, a change in the rotation leads to an instantaneous change in the position of the object, which again influences the spotlight. Overall, we obtain an instantaneous graph of eight edges between the seven multidimensional causal variables. Since the dataset is visually complex, we use the normalizing flow variant of iCITRIS and CITRIS applied on a pretrained autoencoder.

Table 1 shows that iCITRIS-ENCO disentangles the causal variables well and recovers most instantaneous relations in this challenging setup, with one error on average. The temporal graph had more false positive edges due to minor, additional correlations. On the other hand, iCITRIS-NOTEARS struggles with the graph learning and incorrectly oriented edges during training, underlining the benefit of ENCO as the graph learning method in iCITRIS. The baselines have a significantly higher entanglement of the causal variables and struggle with finding the true causal graph. In summary, iCITRIS-ENCO can identify the causal variables and their instantaneous graph in this visually challenging dataset well.

### 4.3 REAL GAME DYNAMICS: CAUSAL PINBALL

Finally, we consider a simplified version of the game Pinball, which naturally comes with instantaneous causal effects: if the user activates the paddles when the ball is close, the ball is accelerated immediately. Similarly, when the ball hits a bumper, its light turns on and the score increases directly. This results in instantaneous effects under common frame rates. In this environment, we consider five causal variables: the position of the left paddle, the right paddle, the ball (position and velocity), the state of the bumpers, and the score. Pinball is closer to a real-world environment than the other two datasets and has two characteristic differences: (1) many aspects of the environment are deterministic, *e.g.* the ball movement, and (2) the instantaneous effects are sparse, *e.g.* the paddles do not influence the ball if it is far away of them. The first aspect violates assumptions of iCITRIS, questioning whether iCITRIS yet empirically works here.

The results in Table 2 suggest that iCITRIS still works well on this environment. Besides disentangling the causal variables well, iCITRIS-ENCO identifies the instantaneous causal graph with minor errors. Interestingly, CITRIS obtains a good disentanglement score as well, which is due to the instantaneous effects being often sparse. Yet, there is still a gap between iCITRIS-ENCO and CITRIS in the instantaneous SHD, showing the benefit of learning the instantaneous graph jointly with the causal variables.

## 5 CONCLUSION

We propose iCITRIS, a causal representation learning framework for temporal intervened sequences with instantaneous effects. iCITRIS identifies the minimal causal variables while jointly learning the causal graph, including the instantaneous relations. In experiments, iCITRIS accurately recovers the causal factors and their graph in two video datasets. While we envision a future application of methods similar to iCITRIS in a reinforcement learning setting, the limiting factor currently are the assumptions on the availability of perfect interventions with known targets. Future work includes investigating a setup where a sequence of actions is needed to perform targeted interventions. Finally, iCITRIS is limited to acyclic graphs, while for instantaneous causal effects cycles could occur under low frame rates, which is also an interesting future direction.

## Author Contributions

P. Lippe conceived the idea, implemented the models and datasets, ran the experiments, and wrote most of the paper. S. Magliacane, S. Löwe, Y. M. Asano, T. Cohen, E. Gavves advised during the project. S. Magliacane co-wrote Sec. 1 and 2. S. Magliacane, S. Löwe, Y. M. Asano, E. Gavves provided comments and feedback throughout the writing process.

## Acknowledgements

We thank Johann Brehmer and Pim de Haan for valuable discussions throughout the project. We also thank SURFsara for the support in using the Lisa Compute Cluster. This work is financially supported by Qualcomm Technologies Inc., the University of Amsterdam and the allowance Top consortia for Knowledge and Innovation (TKIs) from the Netherlands Ministry of Economic Affairs and Climate Policy.

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

# A PROOFS

In this section, we provide the proof for the identifiability theorem 2.2 in Section 2 and Lemma 2.1. The section is structured into three main parts. First, in Appendix A.1, we give an overview of the notation and elements that are used in the proof. Next, we discuss the assumptions needed for Theorem 2.2, with a focus on why they are needed and what a violation of these assumptions can cause. Additionally, we provide a proof of Lemma 2.1 in this subsection. Finally, we provide the proof of Theorem 2.2, structured into multiple subsections as different main steps of the proof. A detailed overview of the proof is provided in Appendix A.3.

## A.1 PRELIMINARIES

To clarify the used notation and definitions in the proof, we first discuss the definitions of the properties of causal models. Next, we review the used notation for all other elements in the proof.

### A.1.1 Causal model definition

Given a causal graph $\mathcal{G} = (V, E)$, each node $i \in V$ is associated with a causal variable $C_i$, which can be scalar or vector valued. Each edge $(i, j) \in E$ represents a causal relation from $C_i$ to $C_j$: $C_i \rightarrow C_j$, where $C_i$ is a *parent* of $C_j$ and $\text{pa}_G(C_i)$ are all parents of $C_i$ in $\mathcal{G}$. We can cluster multiple causal variables $C_1, \ldots, C_K$ in a single variable $C = (C_1, \ldots, C_K)$. $C$ then inherits all incoming and outgoing edges from its components $C_i$ for $i = 1, \ldots, K$. We assume that the underlying latent causal process is a dynamic Bayesian network (DBN) (Dean and Kanazawa, 1989; Murphy, 2002) $G$ over $(C_1, C_2, ..., C_K)$ that is first-order Markov, stationary, and without instantaneous effects. This means that in $G$ each causal factor $C_i$ is instantiated at each time step $t$, denoted by $C_i^t$, and its causal parents can only be causal factors at time $t - 1$, denoted as $C_j^{t-1}$, including its own previous value $C_i^{t-1}$, and factors at time $t$, excluding its own value. In other words, for $t = 1, \ldots, T$ and for each causal factor $i = 1, \ldots, K$ we can model $C_i^t = f_i(\text{pa}_G(C_i^t), \epsilon_i)$, where $\text{pa}_G(C_i^t) \subseteq \{C_1^{t-1}, \ldots, C_K^{t-1}, C_1^t, \ldots, C_{i-1}^t, C_{i+1}^t, \ldots, C_K^t\}$. We also assume all $\epsilon_i$ for $i = 1, \ldots, K$ are mutually independent noises. To represent a DBN, the graph structure must be acyclic. This means that there does not exist a directed path in $\mathcal{G}$ from any node $C_i^t$ back to itself. Further, the structure

of the graph is time-invariant, *i.e.*, $\text{pa}_G(C_i^t) = \text{pa}_G(C_i^1)$ for any $t = 1, \ldots, T$.

We use a binary intervention vector $I^t \in \{0, 1\}^K$ to indicate that a variable $C_i^t$ in $G$ is intervened upon if and only if $I_i^t = 1$. We consider that the intervention vector components $I_i^t$ might be confounded by another $I_j^t, i \neq j$, and represent these dependencies with an unobserved regime variable $R^t$, which is similar to "policy variables" in Spirtes et al. (2000) or "regime indicators" in Didelez et al. (2006); Mooij et al. (2020). We augment the underlying causal graph $\mathcal{G}$ with the intervention variable $I_i^t$ associated with each causal factor $C_i^t$ by including it in its parent set: $\text{pa}_{G'}(C_i^t) = \text{pa}_G(C_i^t) \cup \{I_i^t\}$. Each intervened variable $C_j^t$ in a time step will have its corresponding intervention variable $I_j^t$ set to 1, otherwise the intervention variable will be set to 0 if the causal factor is not intervened upon at the time step $t$. We say that a distribution $p$ is *Markov* w.r.t. the augmented DAG $G'$ if it factors as $p(V') = \prod_{j \in V'} p(V_j \mid \text{pa}_{G'}(V_j))$, where $V_j$ includes the causal factors $C_i^t$, the intervention vector components $I_i^t$, and the regime $R^t$. Moreover, we say that $p$ is *faithful* to a causal graph $G'$, if there are no additional conditional independencies to the d-separations one can read from the graph $G'$. The augmented graph $G'$ can model interventions with an arbitrary number of targets, including observational data. In this paper, we consider *soft* interventions (Eberhardt, 2007), in which the conditional distribution changes while potentially maintaining parent relations, *i.e.*, $p(C_i^t|\text{pa}_G(C_i^t), I_i^t = 1) \neq p(C_i^t|\text{pa}_G(C_i^t), I_i^t = 0)$, and *perfect* interventions (Pearl, 2009), which change a causal variable independent of its original parents, *i.e.*, $p(C_i^t|\text{pa}_G(C_i^t), I_i^t = 1) = p(C_i^t|I_i^t = 1)$. The latter corresponds to a do-operation $\text{do}(C_i^t = c_i)$ in Pearl (2009), where $c_i$ is randomly sampled from $p(C_i^t|I_i^t = 1)$.

### A.1.2 Notation

Throughout the proof, we will the same notation as used in the main paper, and try to align it as much as possible with Lippe et al. (2022b). As a summary, the notation is:

- We denote the $K$ causal factors in the latent causal dynamical system as $C_1, \ldots, C_K$;
- The dimensions and space of a causal variable is denoted as $C_i \in \mathcal{D}_i^{M_i}$ with $M_i \geq 1$ and let $\mathcal{D}_i$ be $\mathbb{R}$ for continuous variables (*e.g.*, spatial position), $\mathbb{Z}$ for discrete variables (*e.g.*, the score of a player) or mixed;
- We group all causal factors in a single variable $C = (C_1, \ldots, C_K) \in \mathcal{C}$, where $\mathcal{C}$ is the causal factor space $\mathcal{C} = \mathcal{D}_1^{M_1} \times \mathcal{D}_2^{M_2} \times \ldots \times \mathcal{D}_K^{M_K}$;
- The data we base our identifiability on is generated by a latent Dynamic Bayesian network with variables $(C_1^t, C_2^t, \ldots, C_K^t)_{t=1}^T$;
- We assume to know at each time step the binary intervention vector $I^t \in \{0, 1\}^{K+1}$ where $I_i^t = 1$ refers to an intervention on the causal factor $C_i^t$. As a special case $I_0^t = 0$ for all $t$;

- For each causal factor $C_i$, there exists a split $s_i^{\text{var}}(C_i), s_i^{\text{inv}}(C_i)$ such that $s_i^{\text{var}}(C_i)$ represents the variable/manipulable part of $C_i$, while $s_i^{\text{inv}}(C_i)$ represents the invariable part of $C_i$;
- The minimal causal split is defined as the one which only contains the intervention-dependent information in $s_i^{\text{var}}(C_i)$, and everything else in $s_i^{\text{inv}}(C_i)$. This split is denoted by $s_i^{\text{var}^*}(C_i)$ and $s_i^{\text{inv}^*}(C_i)$
- At each time step, we can access observations $x^t, x^{t+1} \in \mathcal{X} \subseteq \mathbb{R}^N$;
- There exist a bijective mapping between observations and causal/noise space, denoted by $h : \mathcal{C} \times \mathcal{E} \to \mathcal{X}$, where $\mathcal{E}$ is the space of the noise variables;
- The noise $E^t \in \mathcal{E}$ at a time step $t$ subsumes all randomness besides the causal model which influences the observations. For example, this could be brightness shifts in Causal3D, or color shifts in the Causal Pinball environment since no causal factor is encoded in brightness and color in these setups respectively. While this setting is quite general, we still require that the values of the causal factors must be identifiable from single observations. Hence, the joint dimensionality of the noise and causal model is limited to the image size.
- For any model learning a latent space, we denote the vector of latent variables by $z^t \in \mathcal{Z} \subseteq \mathbb{R}^M$, where $\mathcal{Z}$ is the latent space of dimension $M \geq \dim(\mathcal{E}) + \dim(\mathcal{C})$;
- In iCITRIS, we learn the inverse of the observation function as $g_\theta : \mathcal{X} \to \mathcal{Z}$;
- In iCITRIS, we learn an assignment from latent dimensions to causal factors, denoted by $\psi : [\![1..M]\!] \to [\![0..K]\!]$;
- The latent variables assigned to each causal factor $C_i$ by $\psi$ are denoted as $z_{\psi_i} = \{z_j | j \in [\![1..M]\!], \psi(j) = i\} = \{g_\theta(x^t)_j | j \in [\![1..M]\!], \psi(j) = i\}$;
- The remaining latent variables that are not assigned to any causal factor are denoted as $z_{\psi_0}$;
- In iCITRIS, we learn a directed, acyclic graph $G = (V, E)$ where $V = \{z_{\psi_i} | i \in [\![0..K]\!]\}$ and the edges represent directed causal relations;
- The graph $G$ induces a parent structure which we denote by $z_{\psi_i^{pa}} = \{z_j | j \in [\![1..M]\!], \psi(j) \in \text{pa}_G(i)\}$ where $\text{pa}_G(0) = \emptyset$, *i.e.* the variables in $z_{\psi_0}$ having no instantaneous parents;
- The parents of a causal variable within the same time step $t + 1$ are denoted by $\text{pa}^{t+1}(C_i^{t+1})$, and the parents of the previous time step $t$ by $\text{pa}^t(C_i^{t+1})$;
- As a special case, we denote the function $g_\theta$ with the parameters $\theta$ that precisely model the inverse of the true observation function, $h^{-1}$, as the disentanglement function $\delta^* : \mathcal{X} \to \tilde{\mathcal{C}} \times \tilde{\mathcal{E}}$ with $\tilde{\mathcal{C}} = \mathcal{D}^{\tilde{M}_1} \times \ldots \times \mathcal{D}^{\tilde{M}_K}$ and $\tilde{M}_i$ being the number of latent dimensions assigned to the causal factor $C_i$ by $\psi^*$. We denote the output of $\delta^*$ for an observation $X$ as $\delta^*(X) = (\tilde{C}_1, \tilde{C}_2, \ldots, \tilde{E})$. The representation of $\delta^*$ as a learnable function is de-

noted by $g_\theta^*$ and $\psi^*$;

- In the following proof, we will use entropy as a measure of information content in a random variable. To be invariant to possible invertible transformations, e.g. scaling by 2, we use the notion of the limiting density of discrete points (LDDP) (Jaynes, 1957, 1968). In contrast to differential entropy, LDDP introduces an *invariant measure* $m(X)$, which can be seen as a reference distribution we measure the entropy of $p(X)$ to. The entropy is thereby defined as:

$$H(X) = -\int p(X) \log \frac{p(X)}{m(X)} dx \qquad (4)$$

In the following proof, we will consider entropy measures over latent and causal variables. For the latent variables, we consider $m(X)$ to be the push-forward distribution of an arbitrary, but fixed distribution in $\mathcal{X}$ (e.g. random Gaussian if $\mathcal{X} = \mathbb{R}^n$) through $g_\theta$. For the causal variables, we consider it to be the push-forward through $h^{-1}$. For more details on LDDP, see Lippe et al. (2022b, Appendix A.1.2) and Jaynes (1957, 1968).

## A.2 ASSUMPTIONS FOR IDENTIFIABILITY

In this section, we provide a detailed discussion of the assumptions of iCITRIS to enable the identification of an underlying causal graph with instantaneous effects. We thereby focus on why these assumptions are necessary, and how a violation of those can lead to scenarios where the causal variables and graph is not identifiable.

### A.2.1 Assumption 1: The interventions on the causal variables are perfect

iCITRIS requires perfect interventions on the causal variables, in order to disentangle the variables in latent space. The perfect interventions are necessary to obtain samples in which the dependencies among the causal variables are broken, as stated in Lemma 2.1 and copied here for completeness:

**Lemma A.1.** *A causal variable $C_i$ cannot always be uniquely identified in iTRIS if $C_i$ has instantaneous parents and no perfect interventions on $C_i$ have been observed.*

*Proof.* To prove this Lemma, it is sufficient to present a counterexample for which a variable $C_i$ cannot be uniquely identified. Consider two random, causal variables $C_1, C_2$ with the causal graph $C_1^t \to C_1^{t+1}, C_2^t \to C_2^{t+1}$. The two causal variables $C_1, C_2$ have therefore no instantaneous relations. Further, consider the (soft-interventional) distributions $p_1(C_1^{t+1}|C_t^1, I_1^{t+1})$ and $p_2(C_2^{t+1}|C_2^1, I_2^{t+1})$ whose form can be arbitrary, but for this example, we choose them

to be Gaussian with constant variance:

$$p_1(C_1^{t+1}|C_1^t, I_1^{t+1}) =$$
$$\begin{cases} \mathcal{N}(C_1^{t+1}|\mu_1(C_1^t), \sigma_1(C_1^t)^2) & \text{if } I_1^{t+1} = 0 \\ \mathcal{N}(C_1^{t+1}|\tilde{\mu}_1(C_2^t), \tilde{\sigma}_1(C_1^t)^2) & \text{if } I_1^{t+1} = 1 \end{cases} \quad (5)$$

$$p_2(C_2^{t+1}|C_2^t, I_2^{t+1}) =$$
$$\begin{cases} \mathcal{N}(C_2^{t+1}|\mu_2(C_2^t), \sigma_2(C_2^t)^2) & \text{if } I_2^{t+1} = 0 \\ \mathcal{N}(C_2^{t+1}|\tilde{\mu}_2(C_2^t), \tilde{\sigma}_2(C_2^t)^2) & \text{if } I_2^{t+1} = 1 \end{cases} \quad (6)$$

where $\mu_1, \tilde{\mu}_1, \mu_2, \tilde{\mu}_2, \sigma_1, \tilde{\sigma}_1, \sigma_2, \tilde{\sigma}_2$ are arbitrary, potentially non-linear functions of $C_1^t$ and $C_2^t$ respectively. Further, to consider the simplest case, suppose that the observation $X^t$ at a time step $t$ are the causal variables themselves, $X^t = [C_1^t, C_2^t]$, and we observe data points of all intervention settings, i.e. $I_i^{t+1} \sim \text{Bernoulli}(q)$ with $0 < q < 1$.

Under this setup, the true generative model follows the distribution:

$$p(X^{t+1}|X^t, I^{t+1}) = p(C_1^{t+1}, C_2^{t+1}|C_1^t, C_2^t, I^{t+1}) \quad (7)$$
$$= p(C_1^{t+1}|C_1^t, C_2^t, I_1^{t+1}, I_2^{t+1}) \cdot \atop p(C_2^{t+1}|C_1^t, C_2^t, I_1^{t+1}, I_2^{t+1}) \quad (8)$$
$$= p_1(C_1^{t+1}|C_1^t, I_1^{t+1}) \cdot \atop p_2(C_2^{t+1}|C_2^t, I_2^{t+1}) \quad (9)$$

where $C_1^{t+1} \perp\!\!\!\perp C_2^{t+1}|X^t, I^{t+1}$. To show that the causal variables are not uniquely identifiable, we need at least one other representation which can achieve the same likelihood as the true generative model under all intervention settings $I^{t+1}$. For this, consider the following distribution:

$$p(X^{t+1}|X^t, I^{t+1}) = p(C_1^{t+1}, C_2^{t+1}|C_1^t, C_2^t, I^{t+1}) \quad (10)$$
$$= p(C_1^{t+1}|C_1^t, C_2^t, I_1^{t+1}, I_2^{t+1}) \cdot \atop p(C_2^{t+1}|C_1^t, C_2^t, C_1^{t+1}, I_1^{t+1}, I_2^{t+1}) \quad (11)$$
$$= p_1(C_1^{t+1}|C_1^t, I_1^{t+1}) \cdot \atop \hat{p}_2(C_1^{t+1} + C_2^{t+1}|C_2^t, C_1^{t+1}, I_2^{t+1}) \quad (12)$$
$$= p_1(\hat{C}_1^{t+1}|C_1^t, I_1^{t+1}) \cdot \atop \hat{p}_2(\hat{C}_2^{t+1}|C_2^t, \hat{C}_1^{t+1}, I_2^{t+1}) \quad (13)$$

with $\hat{C}_1^{t+1} = C_1^{t+1}, \hat{C}_2^{t+1} = C_1^{t+1} + C_2^{t+1}$. Note the additional dependency of $\hat{C}_2^{t+1}$ on $\hat{C}_1^{t+1}$, which is possible in the space of possible causal models with an additional instantaneous causal edge $\hat{C}_1^{t+1} \to \hat{C}_2^{t+1}$. The new distribu-

tion $\hat{p}_2$ is identical to the true distribution, since:

$$\hat{p}_2(C_1^{t+1} + C_2^{t+1}|C_2^t, C_1^{t+1}, I_2^{t+1} = 0) =$$
$$= \mathcal{N}(C_1^{t+1} + C_2^{t+1}|C_1^{t+1} + \mu_2(C_2^t), \sigma_2(C_2^t)^2) \quad (14)$$

$$= \frac{1}{\sqrt{2\pi}\sigma_2(C_2^t)} e^{-\frac{1}{2}\frac{\left(C_1^{t+1} + C_2^{t+1} - (C_1^{t+1} + \mu_2(C_2^t))\right)^2}{\sigma_2(C_2^t)^2}} \quad (15)$$

$$= \frac{1}{\sqrt{2\pi}\sigma_2(C_2^t)} e^{-\frac{1}{2}\frac{\left(C_2^{t+1} - \mu_2(C_2^t)\right)^2}{\sigma_2(C_2^t)^2}} \quad (16)$$

$$= \mathcal{N}(C_2^{t+1}|\mu_2(C_2^t), \sigma_2(C_2^t)^2) \quad (17)$$

$$= p_2(C_2^{t+1}|C_2^t, I_2^{t+1} = 0) \quad (18)$$

Similarly, one can show that $\hat{p}_2(C_1^{t+1} + C_2^{t+1}|C_2^t, C_1^{t+1}, I_2^{t+1} = 1) = p_2(C_2^{t+1}|C_2^t, I_2^{t+1} = 1)$. Hence, the alternative representation $\hat{C}_1^{t+1}, \hat{C}_2^{t+1}$ can model the distribution $p(X^{t+1}|X^t, I^{t+1})$ as well as the true causal model. In conclusion, from the samples alone, we cannot distinguish between the two representation $C_1, C_2$ and $\hat{C}_1, \hat{C}_2$, and the model is therefore not identifiable. $\quad\square$

An alternative example with a non-trivial observation function is visualized in Figure 2, which further underlines the problem.

This shows that with soft interventions, one cannot distinguish between causal relations introduced by the observation function and those that are in the true causal model. Perfect interventions, however, provide an opportunity to do so since if we had known that the intervention on $C_2$ is perfect, the second causal model could not have modeled the correct distribution under $I_2 = 1$. This is since under interventions on a variable, all causal relations to its parents are broken, but only the relations introduced by the encoding function remain. Thus, we can distinguish between the two, allowing us to identify the correct causal model.

While we have shown that soft interventions are not sufficient for finding the instantaneous causal graph, this does not necessarily hold for the temporal relations. For instance, one might have interventions that are perfect within a time step, but keeps the dependency to the previous time step. However, for simplicity, we focus here on fully perfect interventions, and leave this relaxation to future work.

### A.2.2 Assumption 2: Additional variables without interventions are not children of intervened variables

In practice, it may not be feasible to obtain interventions for every causal variable in a complex, dynamical system. Thus, we need to deal with having interventions on only a subgroup of the causal variables.

For those variables, we need to take the assumption that they are not children of the variables, for which we have

observed interventions. The necessity of this assumption becomes clear when considering Appendix A.2.1, in which we have shown that soft interventions are not sufficient to tell apart entanglement introduced by the observation function, and entanglement induced by the causal relations. Coming back to the example of the proof in Appendix A.2.1, suppose that we have only given perfect interventions on $C_1$, and not $C_2$. From interventions on $C_1$ alone, we cannot distinguish between the true causal model and $\hat{C}_1^{t+1} = C_1^{t+1}, \hat{C}_2^{t+1} = C_1^{t+1} + C_2^{t+1}$ ($\hat{C}_1^{t+1} \to \hat{C}_2^{t+1}$ as causal graph), since the intervention on $C_1$ does not affect the entanglement and causal mechanism of $\hat{C}_2^{t+1}$. Thus, it is not possible to identify the causal variables and the respective graph if there may exist instantaneous effects from variables with interventions to those without. However, under the assumption that the variables without interventions have no instantaneous relation with the intervened variables, one can distinguish between the solely passively observed variables and those with perfect interventions, since all entanglement between those within a time step must come from the observation function, not the causal model. Still, among all causal variables without interventions, a disentanglement cannot be guaranteed without further assumptions due to the lack of observed change in their causal mechanism, and a causal graph among those may exist as well.

### A.2.3 Assumption 3: The intervention targets are unique for each causal variable

iCITRIS builds upon interventions to identify the causal variables. The intervention targets are not necessarily independent of each other, but can be confounded. For instance, we could have a setting where we only obtain single-target interventions, or a certain variable $C_i$ can only be jointly intervened upon with another variable $C_j$. In this large space of possible experimental settings, we naturally cannot guarantee identifiability all the time. In particular, we require that intervention targets for the different causal variables are unique:

**Lemma A.2.** *All information that is strictly dependent on the intervention target $I_i^t$, i.e. $s^{\mathrm{var}}(C_i)$ - the minimal causal variable of $C_i$, cannot be disentangled from another causal variable, $C_j$ with $j \neq i$, if their intervention targets are identical: $\forall t, I_i^t = I_j^t$.*

*Proof.* Lippe et al. (2022b) have shown that two causal variables $C_i, C_j$ cannot be disentangled from observational data alone if they follow a Gaussian distribution with equal variance over time. Taking this setup, consider that additionally to observational data, we observe samples where both variables have been intervened upon, $I_i^{t+1} = I_j^{t+1} = 1$. If the interventional distribution of $C_i$ and $C_j$ are both Gaussian with the same variance, we have the same non-identifiability as in the observational case. Since the entanglement axes can

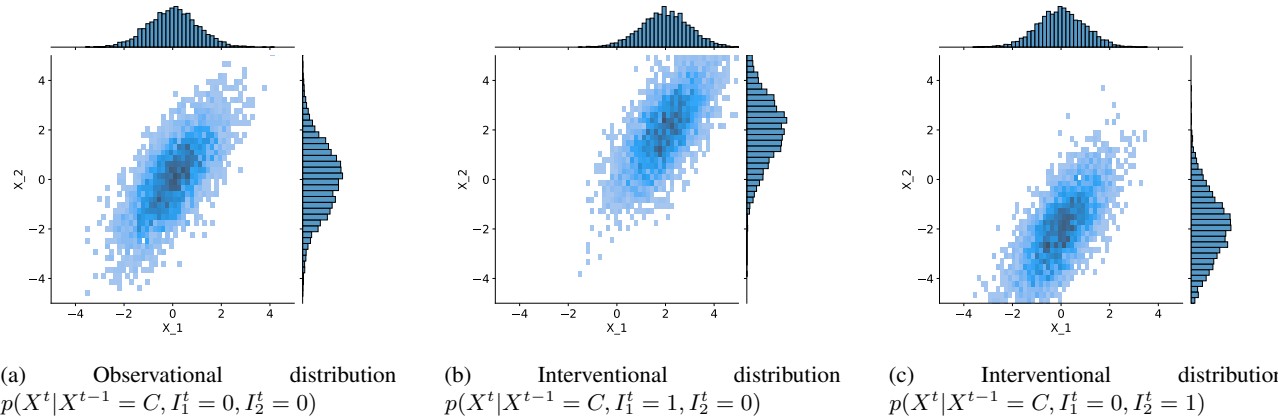

(a)     Observational     distribution $p(X^t|X^{t-1}=C, I_1^t=0, I_2^t=0)$

(b)     Interventional     distribution $p(X^t|X^{t-1}=C, I_1^t=1, I_2^t=0)$

(c)     Interventional     distribution $p(X^t|X^{t-1}=C, I_1^t=0, I_2^t=1)$

Figure 2: Example distribution for showcasing the necessity of perfect interventions for disentangling causal variables with instantaneous effects. Suppose we are given two-dimensional observations $X^t$, for which the observational and interventional distributions are plotted in (a)-(c). The central plot of each subfigure shows a 2D histogram, and the subplots above and on the right show the 1D marginal histograms. For simplicity, we keep the previous time step, $X^{t-1}$, constant here. From the interventional distribution, one might suggest that we have the latent causal graph $C_1 \rightarrow C_2$ since under $I_1^t = 1$, the distribution of both observational distributions change, while $I_2^t = 1$ keeps $X_2$ unchanged. However, the data has been actually generated from two independent causal variables, which have been entangled by having $X^t = [C_1^t, C_1^t + C_2^t]$. We cannot distinguish between these two latent models from interventions that do not reliably break instantaneous causal effects, showing the need for perfect interventions.

transfer between the two setups, $C_i$ and $C_j$ cannot be disentangled, and therefore their minimal causal variables. $\square$

In other words, if two variables are always jointly intervened or passively observed, we cannot distinguish whether information belongs to causal variable $C_i$ or $C_j$. Since the causal system is stationary, having one time step $t$ for which $I_i^t \neq I_j^t$ implies that in the sample limit, we will observe samples with $I_i^t \neq I_j^t$ in the limit as well. Further, when we only observe joint interventions on two variables, $C_i, C_j$, the causal graph among the two variable cannot be identified for arbitrary distributions (Eberhardt, 2007), making the identifiability of the graph and variables impossible.

CITRIS (Lippe et al., 2022b) additionally requires that two intervention targets cannot be the invert of each other, i.e. $I_i^t = 1 - I_j^t$. However, this is not strictly required here, since for $I_i^t = 1$, the perfect interventions imply $C_i$ being independent of all its parents, which is not the case for the observational regime, i.e. $I_i^t = 0$. Thus, as long as $C_i$ and $C_j$ have any parents, there is a possibility to disambiguate the variables even under $\forall t, I_i^t = 1 - I_j^t$.

Still, since there may exist variables without parents, we take the same assumption as Lippe et al. (2022b). Specifically, for every causal variable $C_i$ with observed interventions, we require that the following independence holds:

$$C_i^{t+1} \not\perp\!\!\!\perp I_i^{t+1}|C^t, \text{pa}^{t+1}(C_i^{t+1}), I_j^{t+1} \text{ for any } i \neq j \quad (19)$$

This also implies that there does not exist a variable $C_j$ for which $\forall t, I_i^t = 1 - I_j^t$. As mentioned before, under

additional assumptions such that every causal variable has at least one parents, it can be relaxed to unique interventions.

### A.2.4   Assumption 4: The observational and interventional distributions share the same support

If the observational and interventional distribution do not share the same support, there exist data points for which the intervention targets can be determined from the observation $X^t$ alone. In such situation, the encoder can change its encoding depending on the intervention target, as long as the decoder can yet recover the full observation. This can potentially create representation models that ignore the latent structure, since the intervention targets are already known. Furthermore, when intervention targets are known from seeing causal variables, we potentially introduce new independencies from intervention targets. For instance, if we have the graph $C_1, C_2 \rightarrow C_3$ where $I_3 = 1$ only if $I_1 = 1, I_2 = 0$, we can induce the intervention targets from other causal factors, making $C_3$ essential independent of $I_3$. To prevent such degenerate solutions, we take the assumption that the observational and intervention distributions share the same support. This assumption implies that any data point could come from either the interventional or observational regime, ensuring that the intervention target cannot deterministically be found from an observation $X^t$.

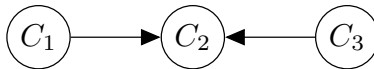

Figure 3: Example instantaneous causal graph between 3 causal variables $C_1, C_2, C_3$. Without temporal dependencies, we could encode information of $C_1$ dependent on $C_3$ without needing an edge in the distribution.

### A.2.5 Assumption 5: Temporal connections and interventions break all symmetries in the distributions

The temporal and interventional dependencies are an essential part in iCITRIS to guarantee identifiability and disentanglement of the causal variables. Without any of these dependencies, there may exist multiple representations that model the same distribution $p(X^t|X^{t-1}, I^t)$, while following the enforced latent structure by iCITRIS. The problem is that variables can functional dependent on each other, where these dependencies exploit symmetries, leaving the distribution unchanged.

For instance, consider the instantaneous causal graph of three variables $C_1, C_2, C_3$ with $C_1, C_3 \rightarrow C_2$, as depicted in Figure 3. Suppose that $C_1$ does not have any temporal parents, and the observational distribution of it follows a Gaussian: $p(C_1^t|I_1^t = 0) = \mathcal{N}(C_1^t|\mu_1, \sigma_1^2)$ with $\mu_1, \sigma_1^2$ being constants. Further, suppose that under interventions, only the standard deviation changes, i.e. $p(C_1^t|I_1^t = 1) = \mathcal{N}(C_1^t|\mu_1, \tilde{\sigma}_1^2)$ with $\tilde{\sigma}_1^2 \neq \sigma_1^2$. Then, for any point $C_1^t = c_1$, there exists a second point, $c_1' = 2\mu_1 - c_1$, which has the same probability for any value of $I_1^t$. This is because both distributions, $p(C_1^t|I_1^t = 0)$ and $p(C_1^t|I_1^t = 1)$, share a symmetry around the mean $\mu_1$.

Now, suppose we have the optimal encoder which maps an observation $X^t$ of this system to the three causal variables with their ground truth values. Then, there exist an alternative encoder, which flips the observed value of $C_1^t$ around the mean $\mu_1$, deterministically conditioned on the remaining variables $C_2^t$ and $C_3^t$. For instance, we could have the following representation $\hat{C}_1^t, \hat{C}_2^t, \hat{C}_3^t$ for the causal variables:

$$\hat{C}_2^t = C_2^t, \hat{C}_3^t = C_3^t, \hat{C}_1^t = \begin{cases} C_1^t & \text{if } \hat{C}_3^t > 0 \\ 2\mu_1 - C_1^t & \text{otherwise} \end{cases} \quad (20)$$

This alternative representation model shares the same likelihood as the optimal encoder in terms of $p(X^t|X^{t-1}, I^t)$, since flipping the value of $C_1^t$ around the mean does not change its probability. Further, despite the flipping, the original observation $X^t$ can be recovered from this alternative representation $\hat{C}^t$ by the decoder, because the possible conditioning factors, i.e. $\hat{C}_3^t$ in this case, are observable to the decoder. Hence, both representations are equally valid for the causal models. Yet, one cannot recover the value of the true causal variable, $C_1^t$, from its alternative representation

$\hat{C}_1^t$ alone, since $\hat{C}_3^t$ needs to be known to invert the example condition. This shows that we can have functional dependencies between representations of causal variables while their distributions remain independent. Thus, there exist more than one representation that cannot be distinguished between from having samples of $p(X^t|X^{t-1}, I^t)$ alone.

More generally speaking, functional dependencies between variables can be introduced if there exists a transformation that leaves the probability of a variable $C_i$ unchanged for any possible value of its parents unseen in $X^t$, i.e. its intervention target $I_i^t$ and temporal parents $C^{t-1}$. Whether this transformation is performed or not can now be conditioned on other variables at time step $t$. Meanwhile, this transformation does not introduce additional dependencies in the causal graph, since the distribution does not change.

To prevent such transformations from being possible, the temporal parents and intervention targets need to break all symmetries in the distributions. We can specify it in the following assumption:

**Assumption 5**: *For a causal variable $C_i$ and its causal mechanism $p(C_i^t|pa^{t+1}(C_i), pa^t(C_i), I_i^t)$, there exist no invertible, smooth transformation $T$ with $T(C_i^t|C_{-i}^t) = \tilde{C}_i^t$ besides the identity, for which the following holds:*

$$\forall C^{t-1}, C^t, I^t : p(C_i^t|pa^{t+1}(C_i^t), pa^t(C_i^t), I_i^t) = \left| \frac{\partial T(C_i^t|C_{-i}^t)}{\partial C_i^t} \right| \cdot p(\tilde{C}_i^t|pa^{t+1}(C_i^t), pa^t(C_i^t), I_i^t) \quad (21)$$

Intuitively, this means that there does not exist any symmetry that is shared across all possible values of the parents (temporal and interventions) of a causal variable. While this might first sound restricting, this assumption will likely hold in most practical scenarios. For instance, if the distribution is a Gaussian, then the assumption holds as long as the mean is not constant since the intervention breaks any parent dependencies are broken by the perfect interventions. The same holds in higher dimensions, as the new symmetries, i.e. rotations, are yet broken if the center point is not constant. Note that these symmetries can be smooth transformations, in contrast to the discontinuous flipping operation on the Gaussian (*i.e.* either we flip the distribution or not, but there is no step in between).

### A.2.6 Assumption 6: Causal graph structure requirements

Besides disentangling and identifying the true causal variables, we are also interested in finding the instantaneous causal graph. This requires us to perform causal discovery, for which we need to take additional assumptions. First, we assume that the causal graph is acyclic, *i.e.* for any causal variable $C_i^t$, there does not exist a path through the directed

causal graph that loops back to it. Note that this excludes different instances over time, meaning that a path from $C_i^t$ to $C_i^{t+\tau}$ is not considered a loop. In real-world setups, there potentially exist instantaneous graphs which are not acyclic, which essentially model a feedback loop over multiple variables. However, to rely on the graph as a distribution factorization, we assume it to be acyclic, and leave extension to cyclic causal graphs for future work. As the second causal graph assumption, we require that the causal graph is faithful, which means that all independences between causal variables are implications of the graph structure, not the specific parameterization of the distributions (Hyttinen et al., 2013; Pearl, 2009). Without faithfulness, the graph might not be fully recoverable. Finally, we assume causal sufficiency, *i.e.* there do not exist any additional latent confounders that introduce dependencies between variables beyond the ones we model. Note that this excludes the potential latent confounder between the intervention targets, and we rather focus on confounders on the causal variables $C_1, ..., C_K$ besides their intervention targets, the previous time step $C^t$, and instantaneous parents $C^{t+1}$.

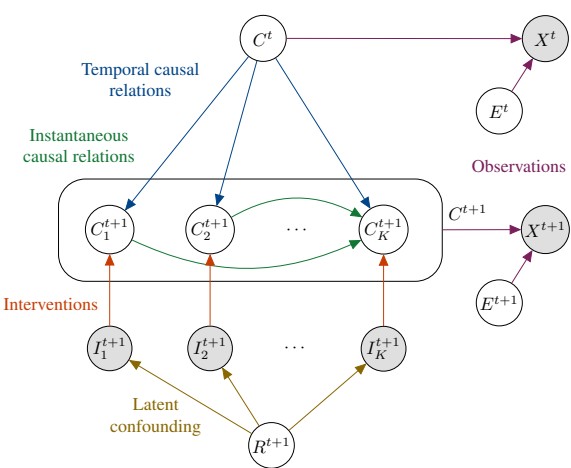

Figure 4: An example causal graph in iTRIS. A latent causal factor $C_i^{t+1}$ can have as potential parents the causal factors at the previous time step $C^t = (C_1^t, ..., C_K^t)$, instantaneous parents $C_j^{t+1}, i \neq j$, and its intervention target $I_i^{t+1}$. All causal variables $C^{t+1}$ and the noise $E^{t+1}$ cause the observation $X^{t+1}$. $R^{t+1}$ is a potential latent confounder between the intervention targets.

## A.3 THEOREM 2.2 - PROOF OUTLINE

The goal of this section is to proof Theorem 2.2: the global optimum of iCITRIS will identify the minimal causal variables and their instantaneous causal graph. The proof follows a similar structure as Lippe et al. (2022b) used for proofing the identifiability in CITRIS, but requires additional steps to integrate the possible instantaneous relations. In summary, we will take the following steps in the proof:

1. (Appendix A.4) Firstly, we show that the function $\delta^*$ that disentangles the true latent variables $C_1, ..., C_K$ and assigns them to the corresponding sets $z_{\psi_1}, ..., z_{\psi_K}$ constitutes a global, but not necessarily unique, optimum for maximizing the likelihood $p(X^{t+1}|X^t, I^{t+1})$.
2. (Appendix A.5) Next, we characterize the class of disentanglement functions $\Delta^*$ which all represent a global maximum of the likelihood, *i.e.* get the same score as the true disentanglement. We do this by proving that all functions in $\Delta^*$ must disentangle the minimal causal variables.
3. (Appendix A.6) In a third step, we show that based on the disentanglement of the minimal causal variables, the causal graph on these learned representations must contain at least the same edges as in the ground truth graph.
4. (Appendix A.7) Finally, we put all parts together and derive Theorem 2.2.

We will make use of Figure 4 summarizing the temporal causal graph, and the notation introduced in Appendix A.1. For the remainder of the proof, we assume for simplicity of exposition that:

- The invertible map $g_\theta$ and the prior $p_\phi(z^{t+1}|z^t, I^{t+1})$ are sufficiently complex to approximate any possible function and distribution one might consider in iTRIS. In practice, over-parameterized neural networks can approximate most functions with sufficient accuracy.
- The latent dimension size is unlimited, *i.e.* $Z \in \mathbb{R}^\infty$. This is assumed such that there are no limitations on how many latent variables $z_{\psi_i}$ can be used to represent a causal factor $C_i$. To maintain invertibility in $g_\theta$, we assume that dimensions beyond $\dim(\mathcal{C}) + \dim(\mathcal{E})$ can potentially become constants. In practice, however, this is not a limiting factor as long as we can overestimate the dimensions of the causal factors and noise variables.
- The sample size for the provided experimental settings is unlimited. This ensures that dependencies and conditional independencies in the causal graph of Figure 4 transfer to the observed dataset, and no additional relations are introduced by sample biases. In practice, a large sample size is likely to give an accurate enough description of the true distributions.

## A.4 THEOREM 2.2 - PROOF STEP 1: THE TRUE MODEL IS A GLOBAL OPTIMUM OF THE LIKELIHOOD OBJECTIVE

We start the identifiability discussion by proving the following Lemma:

**Lemma A.3.** *The true disentanglement function $\delta^*$ that correctly disentangles the true causal factors $C_1^{t+1}, ..., C_K^{t+1}$ from observations $X^t, X^{t+1}$ using the true $\psi^*$ assignment function on the latent variables $Z^{t+1}$ and the true causal*

*graph $G^*$ is one of the global maxima of the likelihood of $p(X^{t+1}|X^t, I^{t+1})$.*

This lemma ensures that the true model is part of the solution space of maximum likelihood objective on $p(X^{t+1}|X^t, I^{t+1})$.

*Proof.* In order to prove this, we first rewrite the objective in terms of the true causal factors. This can be done by using the causal graph in Figure 4, which represents the true generative model:

$$p(X^t, X^{t+1}, C^t, C^{t+1}, I^{t+1}) = p(X^{t+1}|C^{t+1})\cdot$$
$$\left[\prod_{i=1}^{K} p(C_i^{t+1}|C^t, \mathrm{pa}_G^{t+1}(C_i^{t+1}), I_i^{t+1})\right]\cdot \quad (22)$$
$$p(X^t|C^t)\cdot p(C^t)\cdot p(I^{t+1})$$

The context variable $R^{t+1}$ is subsumed in $p(I^{t+1})$, since it is a confounder between the intervention targets and is independent of all other factors given $I^{t+1}$.

In order to obtain $p(X^{t+1}|X^t, I^{t+1})$ from $p(X^t, X^{t+1}, C^t, C^{t+1}, I^{t+1})$, we need to marginalize out $C^t$ and $C^{t+1}$, and condition the distribution on $X^t$ and $I^{t+1}$:

$$p(X^{t+1}|X^t, I^{t+1}) = \int_{C^{t+1}}\int_{C^t} p(X^{t+1}|C^{t+1})\cdot$$
$$\left[\prod_{i=1}^{K} p(C_i^{t+1}|C^t, \mathrm{pa}_G^{t+1}(C_i^{t+1}), I_i^{t+1})\right]\cdot \quad (23)$$
$$p(C^t|X^t)dC^t dC^{t+1}$$

In the assumptions with respect to the observation function $h$, we have defined $h$ to be bijective, meaning that there exists an inverse $f$ that can identify the causal factors $C^t$ and noise variable $E^t$ from $X^t$. The noise variables thereby represent all the stochasticity in the observation function that is not described by the causal factors. For instance, this can be color shifts, limited observation noise, or similar. However, independent of the noise, the causal factors need to be identifiable. This means that the joint dimensionality of the noise and the causal factor are limited by the image size: $\dim(\mathcal{C}) + \dim(\mathcal{E}) \leq \dim(\mathcal{X})$. The observation function $h$ can represent an invertible map between the two spaces even under $\dim(\mathcal{C}) + \dim(\mathcal{E}) < \dim(\mathcal{X})$, since $\mathcal{X}$ does not necessarily need to be $\mathbb{R}^{\dim(\mathcal{X})}$, but rather a subspace.

Using the invertible map, we can write $p(C^t|X^t) = \delta_{f(X^t)=C^t}$, where $\delta$ is a Dirac delta. We also remove $E^t$ from the conditioning set since it is independent of $X^{t+1}$.

This leads us to:

$$p(X^{t+1}|X^t, I^{t+1}) = \int_{C^{t+1}}$$
$$\left[\prod_{i=1}^{K} p(C_i^{t+1}|C^t, \mathrm{pa}_G^{t+1}(C_i^{t+1}), I_i^{t+1}), I_i^{t+1})\right]\cdot \quad (24)$$
$$p(X^{t+1}|C^{t+1})dC^{t+1}$$

We can use a similar step to relate $X^{t+1}$ with $C^{t+1}$ and $E^{t+1}$. However, since we model a distribution over $X^{t+1}$, we need to respect possible non-volume preserving transformations. Hence, we use the change of variables formula with the Jacobian $J_h = \frac{\partial h(C^{t+1}, E^{t+1})}{\partial C^{t+1}\partial E^{t+1})}$ of the observation function $h$ to obtain:

$$p(X^{t+1}|X^t, I^{t+1}) = |J_h|^{-1}\cdot$$
$$\left[\prod_{i=1}^{K} p(C_i^{t+1}|C^t, \mathrm{pa}_G^{t+1}(C_i^{t+1}), I_i^{t+1})\right]\cdot \quad (25)$$
$$p(E^{t+1})$$

Since Equation (25) is a derivation of the true generative model $p(X^t, X^{t+1}, C^t, C^{t+1}, I^{t+1})$, it constitutes a global optimum of the maximum likelihood. Hence, one cannot achieve higher likelihoods by reparameterizing the causal factors or having a different graph, as long as the graph is directed and acyclic.

In the next step, we relate this maximum likelihood solution to iCITRIS, more specifically, the prior of iCITRIS. For this setting, the learnable, invertible map $g_\theta$ is identical to the inverse of the observation function, $h^{-1}$. In terms of the latent variable prior, we have defined our objective of iCITRIS as:

$$p_\phi\left(z^{t+1}|z^t, I^{t+1}\right) = \prod_{i=0}^{K} p_\phi\left(z_{\psi_i}^{t+1}|z^t, z_{\psi_i^{pa}}^{t+1}, I_i^{t+1}\right) \quad (26)$$

Since we know that $g_\theta^*$ is an invertible function between $\mathcal{X}$ and $\mathcal{Z}$, we know that $z^t$ must include all information of $X^t$. Thus, we can also replace it with $z^t = [C^t, E^t]$, giving us:

$$p_\phi\left(z^{t+1}|C^t, E^t, I^{t+1}\right) = \prod_{i=0}^{K} p_\phi\left(z_{\psi_i}^{t+1}|C^t, E^t, z_{\psi_i^{pa}}^{t+1}, I_i^{t+1}\right) \quad (27)$$

Next, we consider the assignment function $\psi^*$. The optimal assignment function $\psi^*$ assigns sufficient dimensions to each causal factor $C_1, ..., C_K$, such that we can consider $z_{\psi_i^*}^{t+1} = C_i^{t+1}$ for $i = 1, ..., K$. Further, the same graph $G$ is used in the latent space as in the ground truth, except that we additionally condition $z_{\psi_i^*}, i = 1, ..., K$ on $z_{\psi_0^*}$. With that, Equation (27) becomes:

$$p_\phi\left(z^{t+1}|C^t, E^t, I^{t+1}\right) =$$
$$\left[\prod_{i=1}^{K} p_\phi\left(z_{\psi_i^*}^{t+1} = C_i^{t+1}|C^t, z_{\psi_i^{pa}}^{t+1}, z_{\psi_0^*}^{t+1}, I_i^{t+1}\right)\right]\cdot \quad (28)$$
$$p(z_{\psi_0^*}^{t+1}|C^t, E^t)$$

where we remove $E^t$ from the conditioning set for the causal factors, since know that $C^{t+1}$ and $E^{t+1}$ is independent of $E^t$. Now, $z_{\psi_0^*}$ must summarize all information of $z^{t+1}$ which is not modeled in the causal graph. Thus, $z_{\psi_0^*}$ represents the noise variables: $z_{\psi_0^*}^{t+1} = E^{t+1}$.

$$
p_\phi\left(z^{t+1}|C^t, E^t, I^{t+1}\right) = \\
\left[\prod_{i=1}^{K} p_\phi\left(z_{\psi_i^*}^{t+1} = C_i^{t+1}|C^t, z_{\psi_i^{pa}}^{t+1}, z_{\psi_0^*}^{t+1}, I_i^{t+1}\right)\right] \cdot \quad (29) \\
p(z_{\psi_0^*}^{t+1} = E^{t+1}|C^t, E^t)
$$

Finally, by using $g_\theta^*$, we can replace the distribution on $z^{t+1}$ by a distribution on $X^{t+1}$ by the change of variables formula:

$$
p_\phi\left(X^{t+1}|C^t, E^t, I^{t+1}\right) = \left|\frac{\partial g_\theta^*(z^{t+1})}{\partial z^{t+1}}\right| \cdot \\
\left[\prod_{i=1}^{K} p_\phi\left(z_{\psi_i^*}^{t+1} = C_i^{t+1}|C^t, z_{\psi_i^{pa}}^{t+1}, z_{\psi_0^*}^{t+1}, I_i^{t+1}\right)\right] \cdot \quad (30) \\
p(z_{\psi_0^*}^{t+1} = E^{t+1}|C^t, E^t)
$$

We can simplify this distribution by using the independencies of the noise term $E^{t+1}$ in the causal graph of Figure 4:

$$
p_\phi\left(X^{t+1}|C^t, E^t, I^{t+1}\right) = \left|\frac{\partial g_\theta^*(z^{t+1})}{\partial z^{t+1}}\right| \cdot \\
\left[\prod_{i=1}^{K} p_\phi\left(z_{\psi_i^*}^{t+1} = C_i^{t+1}|C^t, z_{\psi_i^{pa}}^{t+1}, I_i^{t+1}\right)\right] \cdot \quad (31) \\
p(z_{\psi_0^*}^{t+1} = E^{t+1})
$$

With this, Equation (31) represents the exact same distribution as Equation (25). Therefore, we have shown that the function $\delta^*$ that disentangles the true latent variables $C_1, ..., C_K$ and assigns them to the corresponding sets $z_{\psi_1}, ..., z_{\psi_K}$ constitutes a global optimum for maximizing the likelihood. However, this solution is not necessarily unique, and additional optima may exist. In the next steps of the proof, we will discuss the class of disentanglement functions and graphs that lead to the same optimum. $\square$

## A.5   THEOREM 2.2 - PROOF STEP 2: CHARACTERIZING THE DISENTANGLEMENT CLASS

In this section, we discuss the disentanglement and identifiability results of the causal variables in iCITRIS. We first describe the minimal causal variables in iTRIS, and how they differ to TRIS in CITRIS (Lippe et al., 2022b). Next, we identify the information that must be assigned to individual parts of the latent representation, and similarly, what needs to be disentangled. Finally, we discuss the final setup to ensure disentanglement, including the additional variables in $z_{\psi_0}$.

### A.5.1   Minimal causal variables

Lippe et al. (2022b) introduced the concept of a minimal causal variable as an invertible split of a causal variable $C_i$ into one part that is strictly dependent on the intervention, $s^{\text{var}}(C_i)$, and a part that is independent of it, $s^{\text{inv}}(C_i)$. For iCITRIS, we consider the same concept, but adapt it to the setup of iTRIS.

First, iTRIS assumes the presence of perfect interventions. When given perfect interventions, we can ensure that $s^{\text{inv}}(C_i)$ does not have any parents. This is because under interventions, a causal variable $C_i$ becomes independent of all its parents, and hence $s^{\text{inv}}(C_i)$ must be as well. Since $s^{\text{inv}}(C_i)$ is independent of $I_i$ and thus does not change its mechanism with the intervention, $s^{\text{inv}}(C_i)$ must *always* be independent of all parents of $C_i$. Hence, we can limit our discussion to splits where $s^{\text{inv}}(C_i)$ does not have any parents.

Second, the presence of a causal graph in iCITRIS allows dependencies between different parts of the latent space. Further, $z_{\psi_0}$ can be the parent of any other set of variables, thus allowing for potential dependencies between $s^{\text{inv}}(C_i)$ and $s^{\text{var}}(C_i)$. Note that those, however, must also be cut off by the perfect intervention. Hence, the split $s_i(C_i^t) = (s_i^{\text{var}}(C_i^t), s_i^{\text{inv}}(C_i^t))$ must have the following distribution structure:

$$
p\left(s_i(C_i^{t+1})|C^t, \text{pa}^{t+1}(C_i^{t+1}), I_i^{t+1}\right) = \\
p\left(s_i^{\text{var}}(C_i^{t+1})|C^t, \text{pa}^{t+1}(C_i^{t+1}), s_i^{\text{inv}}(C_i^{t+1}), I_i^{t+1}\right) \cdot \\
p\left(s_i^{\text{inv}}(C_i^{t+1})\right)
$$
$$(32)$$

where

$$
p\left(s_i^{\text{var}}(C_i^{t+1})|C^t, \text{pa}^{t+1}(C_i^{t+1}), s_i^{\text{inv}}(C_i^{t+1}), I_i^{t+1}\right) = \\
\begin{cases}
\tilde{p}\left(s_i^{\text{var}}(C_i^{t+1})\right) & \text{if } I_i^{t+1} = 1 \\
p\left(s_i^{\text{var}}(C_i^{t+1})|\text{pa}(C_i^{t+1}), s_i^{\text{inv}}(C_i^{t+1})\right) & \text{else}
\end{cases}
$$
$$(33)$$

Thereby, the minimal causal variable with respect to its intervention variable $I_i^{t+1}$ is the split $s_i$ which maximizes the information content $H(s_i^{\text{inv}}(C_i^t))$. These relations are visualized in Figure 5.

Causal variables for which the intervention target is constant, i.e. no interventions have been observed, were modeled by $s^{\text{inv}}(C_i) = C_i, s^{\text{var}}(C_i) = \emptyset$ in CITRIS (Lippe et al., 2022b). Here, this does not naturally hold anymore since $s^{\text{inv}}(C_i)$ is restricted to not having any parents. Hence, for the simplicity of exposition in this proof, we add the exception that for a causal variable $C_i$, if $I_i^t = 0$ for all $t$, its minimal causal split is defined as $s^{\text{inv}}(C_i) = C_i, s^{\text{var}}(C_i) = \emptyset$.

### A.5.2   Identifying the minimal causal variables

As a first step towards disentanglement, we postulate the following lemma:

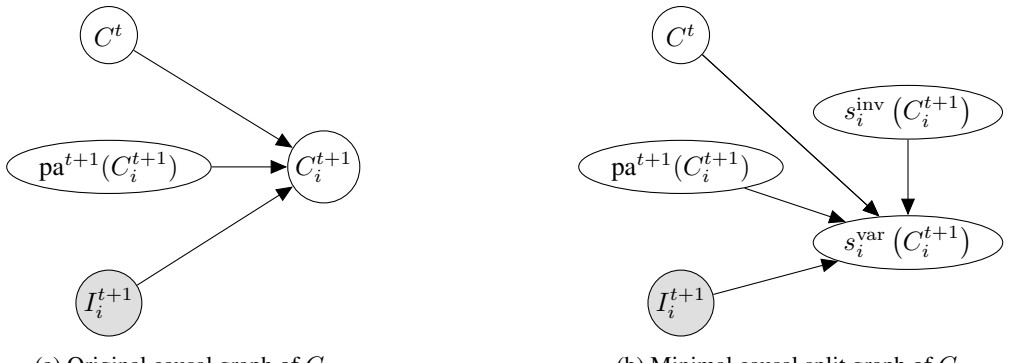

(a) Original causal graph of $C_i$         (b) Minimal causal split graph of $C_i$

Figure 5: The minimal causal variable in terms of a causal graph under iTRIS. (a) In the original causal graph, $C_i^{t+1}$ has as potential parents the causal variables of the previous time step $C^t$ (eventually a subset), its instantaneous parents $\mathrm{pa}^{t+1}(C_i^{t+1})$, and the intervention target $I_i^{t+1}$. (b) The minimal causal variable splits $C_i^{t+1}$ into an invariable part $s_i^{\mathrm{inv}}\left(C_i^{t+1}\right)$ and variable part $s_i^{\mathrm{var}}\left(C_i^{t+1}\right)$. The invariable part $s_i^{\mathrm{inv}}\left(C_i^{t+1}\right)$ is independent of *all* parents due to perfect interventions. However, it can be a parent of $s_i^{\mathrm{var}}\left(C_i^{t+1}\right)$ due to the autoregressive distribution modeling.

**Lemma A.4.** *For all representation functions in the disentanglement class $\Delta^*$, there exist a deterministic map from the latent representation $z_{\psi_i}$ to the minimal causal variable $s^{\mathrm{var}}(C_i)$ for all causal variables $C_i, i = 1, ..., K$.*

This lemma intuitively states that the minimal causal variable $s^{\mathrm{var}}(C_i)$ is modeled in the latent representation $z_{\psi_i}$ for any representation that maximizes the likelihood objective. Note that this does not imply exclusive modeling yet, meaning that $z_{\psi_i}$ can contain more information than just $s^{\mathrm{var}}(C_i)$. We will discuss this aspect in Appendix A.5.3.

*Proof.* In order to prove this lemma, we first review some relations between the conditional and joint entropy. Consider two random variables $A, B$ of arbitrary space and dimension. The conditional entropy between these two random variables is defined as $H(A|B) = H(A, B) - H(B)$ (Cover and Thomas, 2005). Further, the maximum of the joint entropy is the sum of the individual entropy terms, $H(A, B) \leq H(A) + H(B)$ (Cover and Thomas, 2005). Hence, we get that $H(A|B) = H(A, B) - H(B) \leq H(A) + H(B) - H(B) = H(A)$. In other words, the entropy of a random variable $A$ can only become lower when conditioned on any other random variable $B$.

Using this relation, we move now to identifying the minimal causal variables. If a minimal causal variable is the empty set, i.e. $s^{\mathrm{var}}(C_i) = \emptyset$, for instance due to not having observed interventions on $C_i$, the lemma is already true by construction since no information must be modeled in $z_{\psi_i}$. Thus, we can focus on cases where $s^{\mathrm{var}}(C_i) \neq \emptyset$, which implies that $C_i^{t+1} \not\perp I_i^{t+1}$. Therefore, the following inequality must strictly hold:

$$H(C_i^{t+1}|C^t, C_{-i}^{t+1}) < H(C_i^{t+1}|C^t, C_{-i}^{t+1}, I_i^{t+1}) \quad (34)$$

for all $i = 1, ..., K$. Additionally, based on the assump-

tion that the observational and interventional distributions share the same support, we know that the intervention posterior, i.e. $p(I^{t+1}|X^{t+1})$, cannot be deterministic for any data point $X^{t+1}$ and intervention target $I_i^{t+1}$. Thus, we cannot derive $I_i^{t+1}$ from the observation $X^{t+1}$. Thirdly, because every latent variable is only conditioned on exactly one intervention target in iCITRIS and there exist no deterministic function between any pair of intervention targets, one cannot identify $I_i^{t+1}$ in any latent variables except $z_{\psi_i}$. Therefore, the only way in iCITRIS to fully exploit the information of the intervention target $I_i^{t+1}$ is to model its dependent information in $z_{\psi_i}$. As this information corresponds to the minimal causal variable, $s^{\mathrm{var}}(C_i)$, any representation function must model the distribution $p(s^{\mathrm{var}}(C_i)|...)$ in $p(z_{\psi_i}|I_i^{t+1}, ...)$ to achieve the maximum likelihood solution. This is independent of the modeled causal graph structure, meaning that if there exist representation functions with different graphs in $\Delta^*$, then all of them must model $s^{\mathrm{var}}(C_i)$ in $z_{\psi_i}$. Finally, using assumption 5 (Appendix A.2.5), we obtain that this distributional relation implies a functional independence of $s^{\mathrm{var}}(C_i)$ in $z_{\psi_i}$ to any other latent variable. Thus, there exists a deterministic map from $z_{\psi_i}$ to $s^{\mathrm{var}}(C_i)$ in any of the maximum likelihood solutions. $\square$

### A.5.3 Disentangling the minimal causal variables

The previous subsection showed that $z_{\psi_i}$ models the minimal causal variable $s^{\mathrm{var}}(C_i)$. This, however, is not necessarily the only information in $z_{\psi_i}$. For instance, for two random variables $A, B \in \mathbb{R}$, the following distributions are identical:

$$p(A) \cdot p(B|A) = p(A) \cdot p(B + A|A) = p(A) \cdot p(B, A|A) \quad (35)$$

The second distribution can add additional information about $A$ arbitrarily to $B$ without changing the likelihoods.

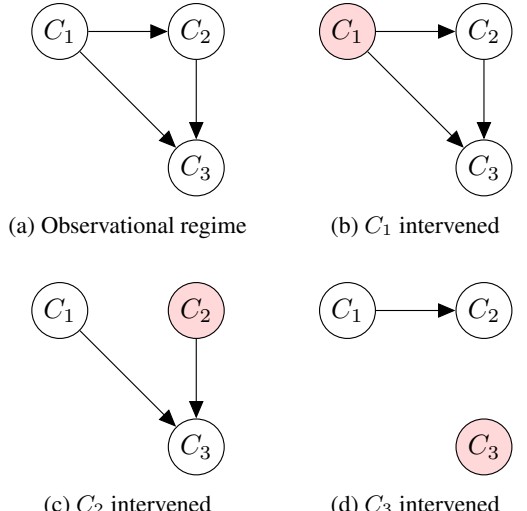

(a) Observational regime      (b) $C_1$ intervened

(c) $C_2$ intervened      (d) $C_3$ intervened

Figure 6: Example instantaneous causal graph between 3 causal variables $C_1, C_2, C_3$, and the augmented graphs under different perfect, single-target interventions. The augmented graphs have the edges to the intervened variables removed. For readability, the intervened variables are colored in red in the graphs.

This is because the distribution is conditioned on $A$, and the conditional entropy of a random variable to itself is $H(A|A) = H(A) - H(A) = 0$. Hence, for arbitrary autoregressive distributions, we cannot disentangle variables from each other purely by looking at the likelihoods.

However, in iTRIS, we are given perfect interventions under which variables are strictly independent of their parents. With this, we postulate the following lemma:

**Lemma A.5.** *For all representation functions in the disentanglement class $\Delta^*$, $z_{\psi_i}$ does not contain information about any other minimal causal variable $s^{\mathrm{var}}(C_j), j \neq i$, except $s^{\mathrm{var}}(C_i)$, i.e. $H(z_{\psi_i}|s^{\mathrm{var}}(C_i)) = H(z_{\psi_i}|s^{\mathrm{var}}(C_i), s^{\mathrm{var}}(C_j))$.*

*Proof.* In order to prove this lemma, we consider all augmented graph structures that are induced by the provided perfect interventions. Specifically, given a graph $G = (V, E)$ with $V$ being its vertices and $E$ its edges, and a set of binary intervention targets $I = \{I_1, ..., I_{|V|}\}$, we construct an augmented DAG $G' = (V', E')$, where $V' = V$ and $E' = E \setminus \{\{\mathrm{pa}_G(V_i) \to V_i\}|i = 1, ..., |V|, I_i = 1\}$. In other words, the augmented graph $G'$ has all its input edges to intervened variables removed. An example for a graph of three variables and its three single-target interventions is shown in Figure 6.

A representation function in the disentanglement class $\Delta^*$ must model the optimal likelihood for *all* intervention-augmented graphs of its originally learned graph $\hat{G}$, since it cannot achieve lower likelihood for any of the graphs than the ground truth. For every pair of variables $C_i, C_j$, assump-

tion 3 (Appendix A.2.3) ensures that there exist one out of three possible experiment sets: (1) we observe $I_i^t = 1, I_j^t = 0$ and $I_i^t = 0, I_j^t = 1$, (2) $I_i^t = 0, I_j^t = 0, I_i^t = 1, I_j^t = 0$, and $I_i^t = 1, I_j^t = 1$, or (3) $I_i^t = 0, I_j^t = 0, I_i^t = 0, I_j^t = 1$, and $I_i^t = 1, I_j^t = 1$. In all cases, there exist at least one augmented graph in which $z_{\psi_i} \perp\!\!\!\perp z_{\psi_j}$ since (2) and (3) observe joint interventions on both variables. In (1), a constant connection between the two variables would require both edges $C_i \to C_j$ and $C_j \to C_i$ in the graph which is not acyclic. Under the augmented graph, where $z_{\psi_i} \perp\!\!\!\perp z_{\psi_j}$, the optimal likelihood can only be achieved if $z_{\psi_i}$ is actually independent of $z_{\psi_j}$, thus not containing any information about $s^{\mathrm{var}}(C_j)$. The same holds for $z_{\psi_j}$. Hence, a representation function in the disentanglement class $\Delta^*$ must disentangle the minimal causal variables in the latent space. $\qquad\square$

### A.5.4 Disentangling the remaining variables

In Appendix A.5.2 and Appendix A.5.3, we have shown that for any solution in the disentanglement class $\Delta^*$, we can ensure that $z_{\psi_i}$ models the minimal causal variable $s^{\mathrm{var}}(C_i)$, and none other. Still, there exist more dimensions that need to be modeled. The causal variables without interventions, the invariant parts of the causal variables, $s^{\mathrm{inv}}(C_i)$, as well as the noise variables $E^t$ are part of the generative model that influence an observation $X^t$. All these variables share the property that they are not instantaneous children of any minimal causal variable, and can only be parents of them. This leads to the situation that any of these variables could be modeled in the latent representation of $z_{\psi_i}$ for an arbitrary $i = 1, ..., K$ as long as $C_i$ is the parent of the same variables. The reason for this is that the distribution modeling of such variables is independent of interventions.

To exclude them from the causal variable modeling, we follow the same strategy as in CITRIS (Lippe et al., 2022b) by taking the representation function that maximizes the entropy of $z_{\psi_0}$:

**Lemma A.6.** *For all representation functions in the disentanglement class $\Delta^*$ that maximize the entropy of $p(z_{\psi_0}|C^t)$, the latent representation $z_{\psi_i}$ models exclusively the minimal causal variable $s^{\mathrm{var}}(C_i)$ for all causal variables $C_i, i = 1, ..., K$.*

*Proof.* Using Lemma A.4 and Lemma A.5, we know that the only remaining information besides the minimal causal variables are the causal variables without interventions, invariant parts of the causal variables, $s^{\mathrm{inv}}(C_i)$, as well as the noise variables $E^t$. All these variables cannot be children of the observed, intervened variables, as the assumption 2 (Appendix A.2.2) states. Thus, the remaining information $\mathcal{M} = \{s^{\mathrm{inv}}(C_1), ..., s^{\mathrm{inv}}(C_K), E^t\}$ can be optimally modeled by $p(\mathcal{M}|z^t)p(z_{\psi_1}, ..., z_{\psi_K}|\mathcal{M}, z^t, I^{t+1})$. This implies that there exist a solution where $z_{\psi_0} = \mathcal{M}$, which can be found by searching for the solution with the maximum en-

tropy of $p(z_{\psi_0}|C^t)$. In this solution, the latent representation $z_{\psi_1}, ..., z_{\psi_K}$ does not model any subset of $\mathcal{M}$, hence modeling the minimal causal variables exclusively. $\qquad\square$

The overall disentanglement result is that we identify the minimal causal variables in $z_{\psi_1}, ..., z_{\psi_K}$, and all remaining information is modeled in $z_{\psi_0}$. Note that the causal variables without interventions, the noise variables and the invariant part of the causal variables can be arbitrarily entangled in $z_{\psi_0}$. Furthermore, since there exist variables in $z_{\psi_0}$ that may not have any temporal parents (e.g. the noise variables and invariable parts of the intervened causal variables), we cannot rely on assumption 5 (Appendix A.2.5) to ensure functional independence. Hence, while the distribution of $p(z_{\psi_0}|z^t)$ is independent of $z_{\psi_1}, ..., z_{\psi_K}$, there may exist dependencies such that for a single data point, a change in $z_{\psi_i}$ can result in a change of the noise or invariable parts of the causal variables in the observational space.

## A.6 THEOREM 2.2 - PROOF STEP 3: IDENTIFIABILITY OF THE CAUSAL GRAPH

In this step of the proof, we discuss the identifiability of the causal graph under the findings of the disentanglement. In the first subsection, we discuss what graph we can optimally find under the disentanglement of the minimal causal variables. In the second part, we then show how the maximum likelihood objective is sufficient for identifying the instantaneous causal graph. Finally, we discuss the identifiability of the temporal causal graph.

### A.6.1 Causal graph on minimal causal variables

The identification of the causal graph naturally depends on the learned latent representations of the causal variables. In Appendix A.5, we have shown that one can only guarantee to find the minimal causal variables in iTRIS. Thus, we are limited to finding the causal graph on the minimal causal variables $s^{\mathrm{var}}(C_1), s^{\mathrm{var}}(C_2), ..., s^{\mathrm{var}}(C_K)$ and the additional variables modeled in $z_{\psi_0}$. The graph between the minimal causal variables is not necessarily equal to the ground truth graph. For instance, consider a 2-dimensional position $(x, y)$ and the color of an object as two causal variables. If the $x$-position causes the color, but the minimal causal variable of the position is only $s^{\mathrm{var}}(C_1) = y$, then the color has only $s^{\mathrm{inv}}(C_1)$ as parent, not $s^{\mathrm{var}}(C_1)$. In the learned graph on the latent representation, it would mean that we do not have an edge between $z_{\psi_1}$ and $z_{\psi_2}$, but instead $z_{\psi_0} \rightarrow z_{\psi_2}$. Hence, we might have a mismatch between the ground truth graph on the full causal variables, and the graph on the modeled minimal causal variables.

Still, there are patterns and guarantees that one can give for how the optimal, learned graph looks like. Due to the interventions being perfect, the invariable part of a causal vari-

able, $s^{\mathrm{inv}}(C_i)$, cannot have any parents. Thus, the parents of a minimal causal variable $s^{\mathrm{var}}(C_i)$ are the same ground truth causal variables as in the true graph, i.e. $\mathrm{pa}(C_i) = \mathrm{pa}(s^{\mathrm{var}}(C_i))$. The difference is how the parents are represented. Since each parent $C_j \in \mathrm{pa}(C_i)$ is split into a variable and invariable part, any combination of the two can represent a parent of $s^{\mathrm{var}}(C_i)$. Thus, the learned set of parents for $s^{\mathrm{var}}(C_i)$, i.e. $\mathrm{pa}(z_{\psi_i})$, must be a subset of $\{s^{\mathrm{var}}(C_j)|C_j \in \mathrm{pa}(C_i)\} \cup \{z_{\psi_0}\}$. This implies that if there is no causal edge between two causal variables $C_i$ and $C_j$ in the ground truth causal graph, then there is also no edge between their minimal causal variables $s^{\mathrm{var}}(C_i)$ and $s^{\mathrm{var}}(C_j)$. The causal graph between the true variables and the minimal causal variables therefore shares a lot of similarities, and in practice, is often almost the same.

The additional latent variables $z_{\psi_0}$ summarize all invariable parts of the intervened variables, the remaining causal variables without interventions, and the noise variables. Therefore, $z_{\psi_0}$ cannot be an instantaneous child of any minimal causal variable, and we can predefine the orientation for those edges in the instantaneous graph.

### A.6.2 Optimizing the maximum likelihood objective uniquely identifies the causal graph under perfect interventions

Next, we can discuss the identifiability guarantees for the graph on the minimal causal variables. For simplicity, we refer to identifying the causal graph on the minimal causal variables as identifying the graph on $C_1, ..., C_K$.

Several causal discovery works have shown before that causal graphs can be identified when given sufficient interventions (Brouillard et al., 2020; Eberhardt, 2007; Lippe et al., 2022a; Pearl, 2009). Since the disentanglement of the causal variables already requires perfect interventions, we can exploit these interventions for learning and identifying the graph as well. In assumption 6 (Appendix A.2.6), we have assumed that the causal graph to identify is faithful. This implies that any dependency between two variables, $C_1, C_2$, which have a causal relation among them ($C_1 \rightarrow C_2$ or $C_2 \rightarrow C_1$), cannot be replaced by conditioning $C_1$ and/or $C_2$ on other variables. In other words, in order to optimize the overall likelihood $p(C_1, ..., C_K)$, we require a graph that has a causal edge between two variables if they are causally related. Now, we are interested in whether we can identify the orientation between every pair of causal variables that have a causal relation in the ground truth graph, which leads us to the following lemma:

**Lemma A.7.** *In iTRIS, the orientation of an instantaneous causal effect between two causal variables $C_i, C_j$ can be identified by solely optimizing the likelihood of $p(C_i, C_j|I_i, I_j)$.*

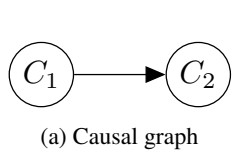

(a) Causal graph

| Exp. | $I_1$ | $I_2$ |
|------|-------|-------|
| $E_0$ | 1 | 0 |
| $E_1$ | 0 | 1 |

(b) Experimental setting 1

| Exp. | $I_1$ | $I_2$ |
|------|-------|-------|
| $E_0$ | 0 | 0 |
| $E_1$ | 1 | 0 |
| $E_2$ | 1 | 1 |

(c) Experimental setting 2

| Exp. | $I_1$ | $I_2$ |
|------|-------|-------|
| $E_0$ | 0 | 0 |
| $E_1$ | 0 | 1 |
| $E_2$ | 1 | 1 |

(d) Experimental setting 3

Figure 7: Identifiability of a causal relation between two variables $C_1, C_2$ under different interventional settings. (a) The causal relation to consider. The discussion is identical in case of the reverse orientation by switching the variable names $C_1$ and $C_2$. (b-d) The tables describe the minimal sets of experiments, i.e. unique combinations of $I_1, I_2$ in the dataset, that guarantee the intervention targets to be unique, i.e. not $\forall t, I_1^t = I_2^t$. Under each of these sets of experiments, we show that the maximum likelihood solution of $p(C_1, C_2 | I_1, I_2)$ uniquely identifies the causal orientation.

Table 3: The probability distribution $p(C_1, C_2 | I_1, I_2)$ for all possible causal graphs among the two causal variables $C_1, C_2$ under different experimental settings. Observational distributions are denoted with $p(...)$, and interventional with $\tilde{p}(...)$. Note that under interventions, it is enforced that $\tilde{p}(...)$ is not conditioned on any parents, since we assume perfect interventions.

| Interventions | | Causal graph | | |
|---|---|---|---|---|
| $I_1$ | $I_2$ | $C_1 \to C_2$ | $C_2 \to C_1$ | $C_1 \perp\!\!\!\perp C_2$ |
| 0 | 0 | $p(C_1)p(C_2\|C_1)$ | $p(C_2)p(C_1\|C_2)$ | $p(C_1)p(C_2)$ |
| 1 | 0 | $\tilde{p}(C_1)p(C_2\|C_1)$ | $p(C_2)\tilde{p}(C_1)$ | $\tilde{p}(C_1)p(C_2)$ |
| 0 | 1 | $p(C_1)\tilde{p}(C_2)$ | $\tilde{p}(C_2)p(C_1\|C_2)$ | $p(C_1)\tilde{p}(C_2)$ |
| 1 | 1 | $\tilde{p}(C_1)\tilde{p}(C_2)$ | $\tilde{p}(C_1)\tilde{p}(C_2)$ | $\tilde{p}(C_1)\tilde{p}(C_2)$ |

*Proof.* To discuss the identifiability of the causal direction between two variables $C_1, C_2$, we need to consider all possible minimal sets of experiments that fulfill the intervention setup in assumption 3 (Appendix A.2.3). These three sets are shown in Figure 7. For all three sets, we have to show that the maximum likelihood of the conditional distribution $p(C_1, C_2 | I_1, I_2)$ can only be achieved by modeling the correct orientation, here $C_1 \to C_2$. For cases where the true graph is $C_2 \to C_1$, the same argumentation holds, just with the variables names $C_1$ and $C_2$ swapped. As an overview, Table 3 shows the distribution $p(C_1, C_2 | I_1, I_2)$ under all possible experiments and causal graphs.

**Experimental setting 1** (Figure 7b) In the first experimental setting, we are given single target interventions on $C_1$ and $C_2$. In the experiment $E_0$ which represents interventions on

$C_1$ and passive observations on $C_2$, the dependency between $C_1$ and $C_2$ persists in the ground truth, i.e. $C_1 \not\perp\!\!\!\perp C_2 | I_1 = 1, I_2 = 0$. Hence, only causal graphs that condition $C_2$ on $C_1$ under interventions on $C_1$ can achieve the maximum likelihood in $E_0$. From Table 3, we see that the only causal graph that does this is $C_1 \to C_2$. Thus, when single-target interventions on $C_1$ are observed, we can uniquely identify the orientation of its outgoing edges.

**Experimental setting 2** (Figure 7c) The second experimental setting provides the observational regime ($E_0$), interventions on $C_1$ with $C_2$ being passively observed ($E_1$), and joint interventions on $C_1$ and $C_2$ ($E_2$). Since the experiment $E_1$ gives us the same setup as in experimental setting 1, we can directly conclude that the causal orientation $C_1 \to C_2$ is yet again identifiable.

**Experimental setting 3** (Figure 7d) In the final experimental setting, $C_1$ is only observed to be jointly intervened upon with $C_2$, not allowing for the same argument as in the experimental settings 1 and 2. However, the causal graph yet remains identifiable because of the following reasons. Firstly, the experiment $E_0$ with its purely observational regime cannot be optimally modeled by a causal graph without an edge between $C_1$ and $C_2$, reducing the set of possible causal graph to $C_1 \to C_2$ and $C_2 \to C_1$. Under the joint interventions $E_2$, both causal graphs model the same distribution. Still, under the experiment $E_1$ where only $C_2$ has been intervened upon, the two distributions differ. The graph with the anti-causal orientation compared to the true graph, $C_2 \to C_1$, uses the same distribution as in the observational regime to model $C_1$, i.e. $p(C_1|C_2)$. In order for this to achieve the same likelihood as the true orientation, it would need to be conditioned on $I_2$ as the following derivation from the true distribution $p(C_1, C_2 | I_1, I_2)$ shows:

$$p(C_1, C_2 | I_1, I_2) = p(C_2 | I_1, I_2) \cdot p(C_1 | C_2, I_1, I_2) \quad (36)$$

$$p(C_1 | C_2, I_1, I_2) = \begin{cases} p(C_1 | I_1) & \text{if } I_2 = 1 \\ p(C_1 | C_2, I_1) & \text{if } I_2 = 0 \end{cases} \quad (37)$$

This derivation shows that $p(C_1 | C_2, I_1, I_2)$ strictly depends on $I_2$ if $p(C_1 | C_2, I_1, I_2 = 1) \neq p(C_1 | C_2, I_1, I_2 = 0)$, which is ensured by $C_1, C_2$ not being conditionally independent in the ground truth graph. As the causal graph $C_2 \to C_1$ models $C_1$ independently of $I_2$, it therefore cannot achieve the maximum likelihood solution in this experimental settings. Hence, the only graph achieving the maximum likelihood solution is $C_1 \to C_2$, such that the orientation can again be uniquely identified.

All other, possible experimental settings must contain one of the three previously discussed experiments as a subset, due to assumption 3 (Appendix A.2.3). Hence, we have shown that for all valid experimental settings, optimizing the maximum likelihood objective uniquely identifies the causal orientations between pairs of variables under perfect interventions. $\quad\square$

Based on these orientations, we can exclude all additional edges that could introduce a cycle in the graph, since we strictly require an acyclic graph. The only remaining non-identified parts of the graph are edges among variables that are independent, conditioned on their parents. In terms of maximum likelihood, these edges do not influence the objective since for two variables $C_1, C_2$ with $C_1 \perp\!\!\!\perp C_2$, $p(C_1) \cdot p(C_2) = p(C_1|C_2) \cdot p(C_2) = p(C_1) \cdot p(C_2|C_1)$. Hence, the equivalence class in terms of maximum likelihood includes all graphs that at least contain the true edges, and are acyclic. By requiring structural minimality, i.e. taking the graph with the least amount of edges that yet fully describe the probability distribution, we can therefore identify the full causal graph between $C_1, ..., C_K$.

### A.6.3 Identifying the temporal causal relations by pruning edges

So far, we have shown that the instantaneous causal relations can be identified between the minimal causal variables. Besides the instantaneous graph, there also exist temporal relations between $C^t$ and $C^{t+1}$, which we also aim to identify:

**Lemma A.8.** *In iTRIS, the temporal causal graph between the minimal causal variables can be identified by removing the edge between any pair of variables $z^t_{\psi_i}, z^{t+1}_{\psi_j}$ with $i, j \in [\![0..K]\!]$, if $z^t_{\psi_i} \perp\!\!\!\perp z^{t+1}_{\psi_j} | z^t_{\psi_{-i}}, pa^{t+1}(z^{t+1}_{\psi_j})$.*

*Proof.* The prior in Equation (1) conditions the latents variables $z^{t+1}$ on all variables of the previous time step, $z^t$. Thus, this corresponds to modeling a fully connected graph from $z^t_{\psi_0}, z^t_{\psi_1}, ..., z^t_{\psi_K}$ to $z^{t+1}_{\psi_0}, z^{t+1}_{\psi_1}, ..., z^{t+1}_{\psi_K}$. Since any temporal edge must be oriented from $z^t$ to $z^{t+1}$, it is clear that the true temporal graph, $G_T$, must be a subset of this graph. Further, since in assumption 6 (Appendix A.2.6), we have stated that the true causal model is faithful, we know that two variables, $z^t_{\psi_i}$ and $z^{t+1}_{\psi_j}$, are only connected by an edge, if they are not conditionally independent of each other: $z^t_{\psi_i} \not\perp\!\!\!\perp z^{t+1}_{\psi_j} | z^t_{\psi_{-i}}, pa^{t+1}(z^{t+1}_{\psi_j})$. This implies that all redundant edges must be between two, conditionally independent variables with: $z^t_{\psi_i} \perp\!\!\!\perp z^{t+1}_{\psi_j} | pa^t(z^{t+1}_{\psi_j}), pa^{t+1}(z^{t+1}_{\psi_j})$ with $pa^t(z^{t+1}_{\psi_j})$ being a subset of $z^t_{\psi_{-i}}$. Thus, we can find the true temporal graph by iterating through all pairs of variables $z^t_{\psi_i}$ and $z^{t+1}_{\psi_j}$, and remove the edge if both of them are conditionally independent given $z^t_{\psi_{-i}}, pa^{t+1}(z^{t+1}_{\psi_j})$. $\qquad\square$

### A.7 THEOREM 2.2 - PROOF STEP 4: FINAL IDENTIFIABILITY RESULT

Using the results derived in Appendix A.4, Appendix A.5 and Appendix A.6, we are finally able to derive the full identifiability results. In Appendix A.5, we have shown that any solution that maximizes the likelihood $p_{\phi,\theta,G}(x^{t+1}|x^t, I^{t+1})$ disentangles the minimal causal vari-

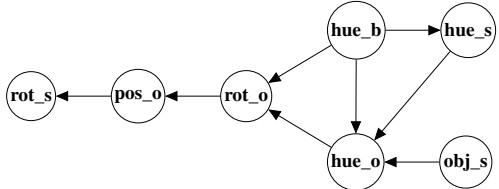

Figure 8: The instantaneous causal graph in the Instantaneous Temporal Causal3DIdent dataset. The graph contains several common sub-structures, such as a chain (rot_o→pos_o→rot_s), a fork (hue_o,hue_b→rot_o), and confounders (hue_b→hue_s,hue_o). The most difficult edges to recover include rot_o→pos_o since the object orientation has a complex, non-linear relation to the observation space which is difficult to model and prone to noise. Further, the edge hue_b,hue_s→hue_o only holds for two object shapes (Hare and Dragon), for which the background and spotlight hue have an influence on the object color. For the other five object shapes, the object color is independent of the other two parents.

ables of $C_1, ..., C_K$ in $z_{\psi_1}, ..., z_{\psi_K}$. Further, we are able to summarize all remaining variables in $z_{\psi_0}$ by maximizing the entropy (LDDP) of $p_\phi(z^{t+1}_{\psi_0}|z^t)$. In Appendix A.6, we have used this disentanglement condition to show that the causal graph that maximizes the likelihood must have at least the same edges as the ground truth graph on the minimal causal variables. To obtain the full ground truth graph, we need to pick the one with the least edges.

These aspects together can be summarized into the following theorem:

**Theorem A.9.** *Suppose that $\phi^*$, $\theta^*$, $\psi^*$ and $G^*$ are the parameters that, under the constraint of maximizing the likelihood $p_{\phi,\theta,G}(x^{t+1}|x^t, I^{t+1})$, maximize the information content of $p_\phi(z^{t+1}_{\psi_0}|z^t)$ and minimize the number of edges in $G^*$. Then, with sufficient latent dimensions, the model $\phi^*, \theta^*, \psi^*$ learns a latent structure where $z^{t+1}_{\psi_i}$ models the minimal causal variable of $C_i$, and $G^*$ is the true instantaneous causal graph between these minimal causal variables. Further, pruning edges between time steps $t$ and $t + 1$ identifies the true temporal graph. Finally, $z_{\psi_0}$ models all remaining information.*

## B DATASETS

The following section gives a detailed overview of the datasets. Appendix B.1 discusses the Instantaneous Temporal Causal3DIdent dataset, and Appendix B.2 the Causal Pinball dataset.

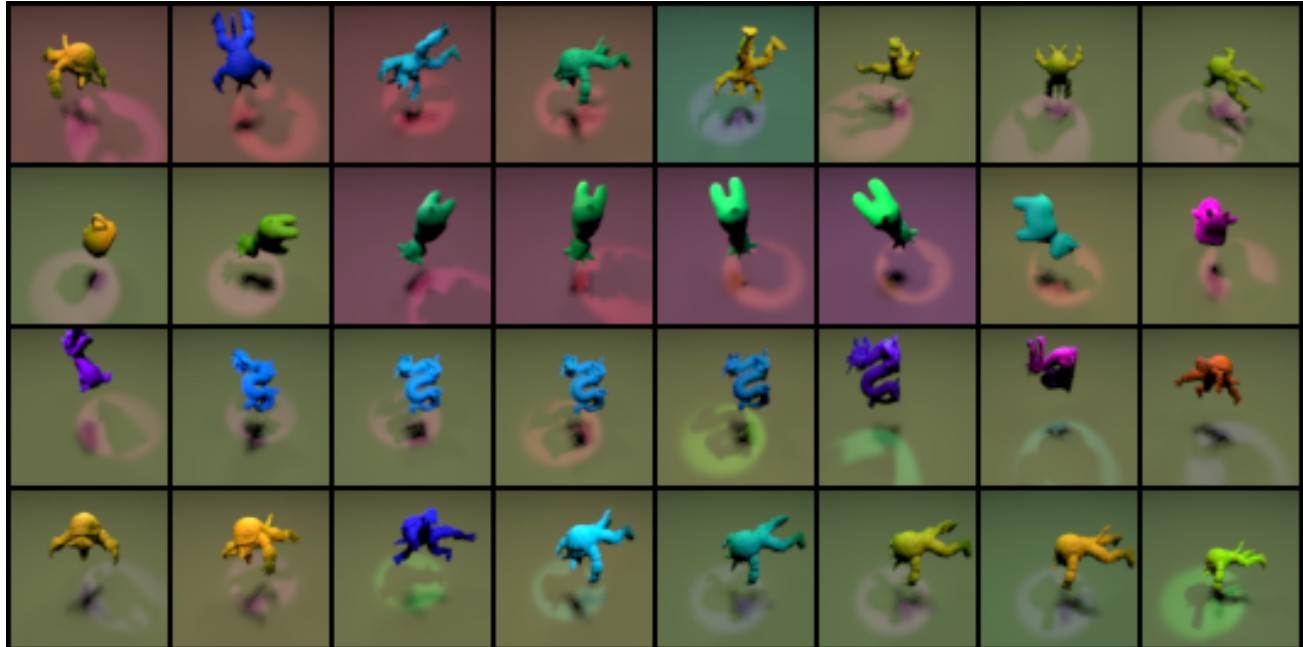

Figure 9: Example sequence from the training set of the Instantaneous Temporal Causal3DIdent dataset (from left to right, top to bottom). Each image is of size $64 \times 64$ pixels. One can see the instantaneous effects of the background influencing the object color, for instance, or the object color again influencing the rotation of the object.

## B.1 INSTANTANEOUS TEMPORAL CAUSAL3DIDENT

The creation of the Instantaneous Temporal Causal3DIdent dataset closely followed the setup of von Kügelgen et al. (2021); Lippe et al. (2022b), and we show an example sequence of the dataset in Figure 9. We used the code provided by Zimmermann et al. (2021)[1] to render the images via Blender (Blender Online Community, 2021), and used the following seven object shapes: Cow (Crane, 2021), Head (Rusinkiewicz et al., 2021), Dragon (Curless and Levoy, 1996), Hare (Turk and Levoy, 1994), Armadillo (Krishnamurthy and Levoy, 1996), Horse (Praun et al., 2000), Teapot (Newell, 1975). As a short recap, the seven causal factors are: the object position as multidimensional vector $[x, y, z] \in [-2, 2]^3$; the object rotation with two dimensions $[\alpha, \beta] \in [0, 2\pi]^2$; the hue of the object, background and spotlight in $[0, 2\pi)$; the spotlight's rotation in $[0, 2\pi)$; and the object shape (categorical with seven values). We refer to Lippe et al. (2022b, Appendix C.1) for the full detailed dataset description of Temporal Causal3DIdent, and describe here the steps taken to adapt the datasets towards instantaneous effects.

The original temporal causal graph of the Temporal Causal3DIdent dataset contains 15 edges, of which 8 are between different variables over time. Those relations form an acyclic graph, which we can directly move to instantaneous relations. Thus, the adjacency matrix of the temporal graph

[1] https://github.com/brendel-group/cl-ica

is an identity matrix, while the instantaneous causal graph is visualized in Figure 8. The causal mechanisms remain unchanged, except that the inputs may now be instantaneous. For instance, the spotlight rotation is adapted as follows:

Previous version:
$$\text{rot\_s}^{t+1} = f\left(\text{atan2}(\text{pos\_x}^t, \text{pos\_y}^t), \text{rot\_s}^t, \epsilon_{rs}^{t+1}\right) \quad (38)$$

Instantaneous version:
$$\text{rot\_s}^{t+1} = f\left(\text{atan2}(\text{pos\_x}^{t+1}, \text{pos\_y}^{t+1}), \text{rot\_s}^t, \epsilon_{rs}^{t+1}\right) \quad (39)$$

where $f(a, b, c) = \frac{a-b}{2} + c$. The causal parents of other variables, here pos_x and pos_y, are now instantaneous instead of the previous time step. Hence, an intervention on the position will lead to an instantaneous effect on the rotation of the spotlight. All remaining aspects of the dataset generation are identical to the Temporal Causal3DIdent dataset.

## B.2 CAUSAL PINBALL

The Causal Pinball dataset is a simplified environment of the popular game Pinball, as shown in Figure 10. In Pinball, the user controls two fixed paddles on the bottom of the playing field, and tries to hit the ball such that it collides with various objects for scoring points. There are several versions of Pinball, but for this dataset, we limit it to the essential parts representing the five, multidimensional causal variables:

- The **ball** is defined by four dimensions: the position on

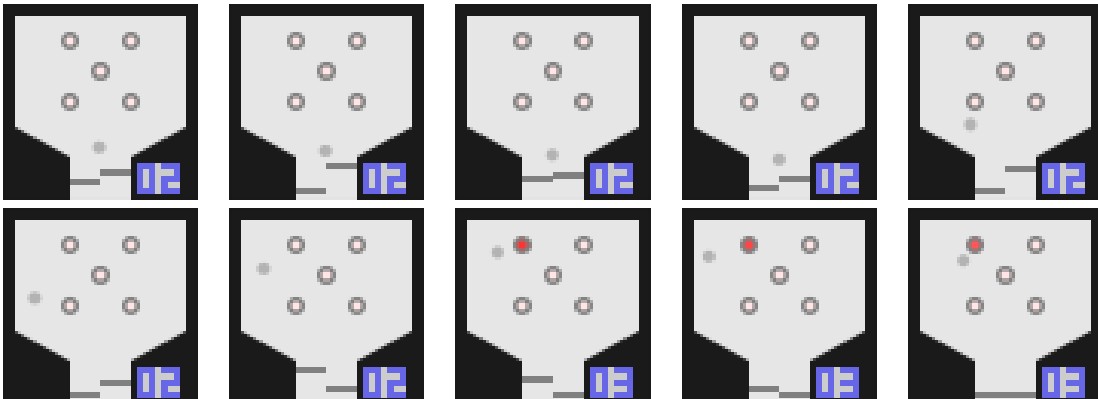

Figure 10: An example sequence of the Pinball dataset, from left to right, top to bottom. The paddles, *i.e.* the two gray rectangles in the bottom center, are accelerated forwards under interventions such that they make a large jump within an image. For instance, in image 5, the right paddle has been intervened upon and hits the ball (gray circle). It is accelerated immediately, showcasing the instantaneous effect between the two. When no interventions on the paddles are given, they slowly move backwards. In image 8, the ball hits a bumper (5 circle centers with light red filling) which lights up. This represents the scoring of a point, as the instantaneous increase in points shows in image 8 (the digits in the bottom right corner). Note that technically, there is no winning or losing state here since we do not focus on learning a policy, but instead a causal representation of the components. Further, not shown here, there exist a fourth channel representing the ball's velocity.

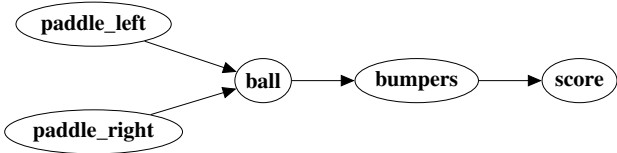

Figure 11: The instantaneous causal graph in the Causal Pinball dataset. An intervention on the paddles can have an immediate effect on the ball by changing its position and velocity. A change in the ball's position again influences the bumpers, whether their light is activated or not. Finally, when the bumpers are activated, the score increases in the same time step.

the x- and y-axis, and its velocity in x and y. Both are continuous values, with the position being limited to the available spots on the field.

- The **left paddle y-position** (paddle_left) describes the position of the left paddle. Its maximum is close-to the top of the black border next to it (*e.g.* image 7 in Figure 10), and its minimum is close to the bottom (*e.g.* image 10 in Figure 10).
- The **right paddle y-position** (paddle_right) is similar to paddle_left, just for the right paddle.
- The **bumpers** represent the activation, *i.e.* the light, of all 5 bumpers. It is a five-dimensional continuous variable, each dimension being between 0 (light off, *e.g.* image 1 in Figure 10) and 1 (light fully on, *e.g.* image 8 in Figure 10).
- The **score** is a categorical variable summarizing the number of points the player has scored. Its value ranges from 0 to a maximum of 20.

The dynamics between these causal factors resembles the standard game dynamics of Pinball, which results in the instantaneous causal graph in Figure 11. The ball can collide with the paddles, borders, and bumpers. When it collides with the borders, it is simply reflected, and we reduce its velocity by 10% (*i.e.* multiply by 0.9). Under collisions with the paddles, we distinguish between a collision where the paddle has been static or moving backwards, versus a collision where the paddle was moving. When the paddle was static, we use the same collision dynamics as the borders, except that we reduce its $y$-velocity by 70% to reduce oscillations around the paddle position. When the paddle was moving, we instead set the $y$-velocity of the ball to the $y$-velocity of the paddle. Finally, when the ball collides with a bumper, it activates the bumper's light and reflects from it, similar to the borders. When a bumper's light is turned on, we increase the score by one, but include a 5% chance that the score is not increased to introduce some stochastic elements and faulty components in the game. Next to the collisions, the ball is influenced by a gravity towards the bottom, adding a constant every time step to its $y$-velocity, and friction that reduces its velocity by 2% after each time step.

In terms of interventions, we sample the interventions on the five elements independently, but with a chance that would correspond more closely to the game dynamics. Specifically, we intervene on the paddles in 20% of the frames, 10% on the ball, and 5% each the score and bumpers. An intervention of the paddle represents it moving forwards, from its previous position, to a randomly sampled position between the middle and maximum paddle position. Its velocity is set to the difference between the previous position and new position. Since these interventions are usually elements of the

standard Pinball game play, we sample them rather often with 20%. An intervention on the ball represents moving it to a position between the two paddles and the bumpers, with a small velocity sampled randomly. In real-life, this would correspond to a player picking up the ball and placing it in a new position. An intervention on the bumpers is that we randomly set the bumper lights either to 0 or 1 with a 50% chance. Finally, an intervention on the score resets it to a random value between 0 and 4.

To render the images, we use matplotlib (Hunter, 2007) and a resolution of $64 \times 64$ pixels. The images are generated by having a single sequence of 150k images.