# OpenReview forum: "iCITRIS: Causal Representation Learning for Instantaneous Temporal Effects"
_auai.org/UAI/2022/Workshop/CRL — CRL@UAI 2022 Poster_

### Official Review · Reviewer_4Vw2 · 2022-06-27
**A good fit of the workshop. A work with rigorous proofs but lack of essential introduction, further discussion, and more details of the proposed method.**

**Rating:** 7
**Confidence:** 4

**Review:**

The paper aims at reconstructing the causal variables from the temporal sequences of observations in the presence of instantaneous effects. It first shows the difficulty of the task and then provides sufficient assumptions to identify the causal representation, such as the perfect intervention on the latent causal variables. Then it provides the proof of the identification of the causal representation by maximizing likelihood, maximizing the information content, and minimizing the edges of a causal graph.

Pros:

- The rigorous proof of Thm. 2.2 is appreciated and valuable for the community.
- The presentation of the paper is in general well organized, i.e., first pointing out the hardness of the task and then showing a solution.

Cons:

My main concern with the paper is the lack of essential details and discussion about causal representation. Currently, causal representation is still at its early stage, so it would be appreciated if a proper justification, a well-formulated concept/definition, a convincing experimental results of causal representation learning are provided. Unfortunately, given the limited space 6 pages, besides including the hardcore part of the method to show the correctness, it is hard to show and the work is currently lack of the other interesting and valuable points.

Nevertheless, I think that the work fits the value and the purpose of the workshop and that it would be a good fit for the communication and the discussion about causal representation learning.

Here I only listed some of the related points (maybe potentially can be helpful for further submission of the work):

**Introduction:**

- Given that the audience may not be familiar with the concept and the field, at the current stage of its development, it deserves to introduce more about the causal representation learning from temporal sequences, especially what it is and how to make it.
- I realized that the work is highly related to and relied on (Lippe et al. 2022b). For the same reason, the audience is very likely to be unfamiliar with the new work; however, the paper seems to assume that the audience needs to know it. So perhaps introduce more and provide more context about it.

**Sec. 2:**

- **Definition of causal factors**
    - What is the definition of causal factors? And why so?
    - What is the meaning of Mi > 1 ?
- **Causal structure:**
    - Discussion and justification required, e.g., pros and cons of dynamical Bayesian networks, and which type of dynamics can be modeled well and which type cannot?
- **Assumptions**
    - Discussion and justification required, i.e., causal sufficiency: why this is necessary? Different from causal discovery, the method requires perfect intervention. Maybe elaborate that under this setup, why it is still required.
- **Interventions**
    - Discussion and  justification required: whether intervention on the latent variables is reasonable or not
    - How to do it in practice?
    - What if we don’t not know what the latent variables, and how to intervene?

**Presentation of the method:**

- Besides the presentation of the model (Sec. 2.3, Sec. 3.1)and the optimization (Sec. 3.2), it deserves more details and a well-structured presentation of the model and its theoretical results, which can be better and more suitable than the stacking style.

**Some other questions and suggestions:**
- How to identify the dimension of latent variables? What is the consequence of misspecification?
- Given a new setup with the perfect intervention, two interesting points can be considered to be further elaborate:
    - How to design the intervention sequence?
    - What is the learned representation?

---

### Official Review · Reviewer_Jg3n · 2022-07-01
**good paper with clear contributions**

**Rating:** 7
**Confidence:** 4

**Review:**

Summary:
To address the general problem of identifying the causal relations among intervened latent variables from entangled observations, this paper extends the method named iCITRIS, which releases the limitations in CITRIS by allowing instantaneous causal effects. The latent variables are assumed to be affected by the variables values at the previous time and some perfect interventions. The proposed method jointly solves the disentanglement and causal discovery task while obtaining some identifiable results. Experimental results show the effectiveness of the proposed method.

Concerns:
1.	The identifiable results provided in this paper is too weak or not a well-defined identifiability. As claimed by authors, the global minimum for the full likelihood objective is not necessarily unique, which is away from the identifiable results as we usually defined. The authors are recommended to consider re-claim their “identifiable results”. For the disentanglement process, the authors may follow the iVAE-series works to constraint the distributions of latent variables or restrict the nonlinear mixing functions as in Yang et al., ICLR22 (Nonlinear ICA Using Volume-Preserving Transformations.)
2.	The authors take NOTEARS and ENCO as baselines to discovery the causal relations among the latent variables. To my knowledge, NOTEARS also needs strict assumptions to identify the causal structures (linear Gaussian with Equal Variance). Moreover, the NOTEARS + Gumbel-Softmax trick is also a well-known method named  MCSL (Ng et al., SDM22), which is also leveraged to learn the causal relations among latent variables in CausalVAE (Yang et al., CVPR21).
3.	The authors are suggested to provided some simple simulated results to support the consistency of their methods.
4.	The problem addressed in this paper is very interesting and important. The proposed end-to-end framework/method is clear and rather effective according to the experimental results. However, the theoretical contribution is weak and slightly overclaimed.

---

### Meta-Review · Program_Chairs · 2022-07-06

**Recommendation:** Accept (Poster)
**Confidence:** 4

**Metareview:**

Both reviewers agree that the paper is a good addition to the workshop; I therefore recommend acceptance. The authors are encouraged to take the comments of reviewer `Jg3n` (type of identifiability and additional baselines), as well as the detailed suggestions by reviewer `4Vw2` into account.

---

### Decision · Program_Chairs · 2022-07-06

Accept (Poster)